# Generating Physically Sound Designs from Text and a Set of Physical Constraints

**Gregory Barber, Todd C. Henry, Mulugeta A. Haile**
DEVCOM Army Research Laboratory
Aberdeen Proving Ground, MD 21005
`gregory.j.barber2.civ@army.mil`

## Abstract

We present TIDES, a text informed design approach for generating physically sound designs based on a textual description and a set of physical constraints. TIDES jointly optimizes structural (topology) and visual properties. A pre-trained text-image model is used to measure the design's visual alignment with a text prompt and a differentiable physics simulator is used to measure its physical performance. We evaluate TIDES on a series of structural optimization problems operating under different load and support conditions, at different resolutions, and experimentally in the lab by performing the 3-point bending test on 2D beam designs that are extruded and 3D printed. We find that it can jointly optimize the two objectives and return designs that satisfy engineering design requirements (compliance and density) while utilizing features specified by text.

## 1 Introduction

Generative design has advanced rapidly and independently across the text-to-image (physics agnostic) and engineering (physics constrained) design domains. In this work, we bridge the two to address a major challenge with each. First, text-image models [1–6] are powerful pattern generators and understand design concepts such as bilateral symmetry, asymmetry, and shapes such as hexagons, arches, and triangles. However, their application to the domain of physical design is limited due to their inability to consider crucial physical design constraints, such as excessive structural deformation, material weight, and manufacturability, in the generated designs. That is to say, while they can generate artistic depictions of "a chair in the shape of an avocado" they can not ensure the chair can support weight, is made of real materials, or can be constructed. Here we introduce a method for placing physical constraints on the output of a text-image generator ensuring the generated design is physically sound and validate by 3D printing and testing a set of designs in the lab.

Second, structural optimization is an approach for designing load-bearing structures such as support columns, beams, and bridges. Given a set of applied forces, materials, and supports, the goal is to generate the design that offers the greatest resistance to the applied force. However, it is challenging to embed complex design goals into the design process, e.g. specifying the type of structural features present in the design such as arch, hexagonal, or triangular supports or an aesthetic goal. Here, we introduce a method for using text-based guidance to explore the design space and return diverse functional designs that utilize visual features specified by text. That is to say, we modify the structural optimization goal from generating the strongest possible design given a set of physical constraints to generating the strongest possible design that meets an additional visual design criteria supplied by text, e.g. "Generate a chair shaped like an avocado that supports the most weight". Additionally, unrelated text prompts have been found to increase originality in human generated designs [7]. We aim to investigate if a similar phenomenon can be observed in an algorithmic design approach.

39th Conference on Neural Information Processing Systems (NeurIPS 2025).

To address these challenges, we developed the Text Informed DESign (TIDES) approach. TIDES consists of three key components: a design generator, a physics simulator, and a visual judge. The design generator parameterizes the design space and determines what material is placed at each point in the design space. The physics simulator provides the means for evaluating the physical performance of the design, and its selection depends on the specific design task at hand. In the current work, we demonstrate on a fully differentiable structural design problem and use the differentiable finite element solver from [8]. For non-differentiable design tasks such as soft robotics [9] or game level design [10], TIDES can be adapted to use a gradient free or genetic algorithm. The visual judge measures the perceived visual similarity of the design to a textual input. A pretrained CLIP model [11] is used as the visual judge. The goal of optimization in TIDES is to jointly maximize the perceived visual similarity of the design to a textual input and the physical performance of the design according to a selected performance metric such as structural stiffness and material cost.

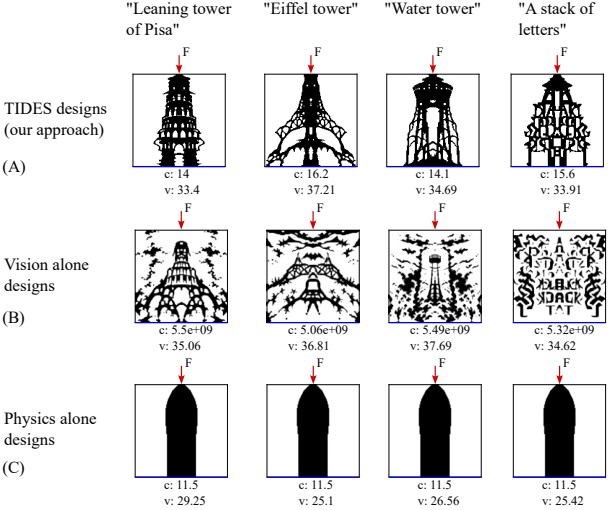

Figure 1: Tower design problem. The red arrow indicates the location a force is applied, and the blue line indicates the support the design rests on. The visual performance (v) of each design is given by the CLIP score. The physical performance is measured by the compliance (c), low compliance indicates a greater resistance to the force. (A) Designs generated from our TIDES approach display features specified by text and resist the applied force. (B) Designs generated from a vision loss alone have no understanding of physics indicated by floating material and orders of magnitude higher compliance. (C) Designs generated from a physics loss alone are physically sound and display simple solid features.

For example, consider the tower structural optimization problem given in Figure 1. Here a load is applied from the top of the design space and the goal is to place the materials in the design space to carry the load with minimum deformation, i.e. maximize the stiffness of the design by minimizing compliance ($c$). The text prompts specified are: "Leaning tower of Pisa, dark black outline", "Eiffel tower, dark black outline", "Water tower, dark black outline" and "A stack of letters, dark black outline", and the goal is to embed features of these structures into the design. As a point of reference, we include the results for a standard structural optimization, design optimized for physical performance alone in Figure 1C and the results for maximizing a vision only criteria (CLIP score) for a given text prompt in Figure 1B. The physics alone designs utilize a simple solid support structure that does not resemble any of the text prompts, indicated by the low clip scores ($v$). The vision alone designs display features given by the text prompt, but have no understanding of physics indicated by the absurd compliance values and floating material. In Figure 1A, we plot the results for our TIDES approach for linking the visual and physical properties of the design. Here, TIDES returns designs that utilize complex support features specified by text: the "Leaning tower of Pisa" design utilizes arch supports, the "Eiffel tower" utilizes a truss structure, the "Water tower" utilizes leg supports, and the "stack of letters" contains supports resembling letters, while resisting deformation from the applied force.

In summary our contributions are:

- **Physics constrained text-image generation**: We develop a framework for placing physical constraints on a pretrained text-image generator to ensure the generated design is physically sound and validate by 3D printing designs generated by our approach and testing them under load for a 3-point bending problem. This indicates it is not necessary to train a new text to design model from scratch on physics data to generate sound designs.
- **Text informed co-design**: We embed a visual constraint into structural optimization, allowing complex design goals to be conveyed to the optimizer by text while respecting the underlying physical constraints. This is a step towards co-design with text providing an intuitive means for shaping the structural support features.
- **Generating diverse physically sound designs**: We find that varying the text prompt allows the generation of diverse designs that perform competitively in terms of structural compliance.

## 2 Background and Related Work

### 2.1 Structural optimization

Structural optimization is a specific field within topology optimization [12] concerned with the design of load-bearing structures such as support columns, beams, and bridges. One of the primary goals of structural optimization is to generate a design that can effectively withstand applied forces while minimizing the overall material cost. In this section, we provide a brief introduction to structural optimization. Introductions and tutorials in structural optimization [13, 14] exist for readers unfamiliar with the domain who are looking for a more thorough introduction.

A structural optimization problem is defined by four factors. The first is the design space. The design space bounds the design and is specified by the dimensions of the design problem. For computational purposes this space is divided into a grid of cells called finite elements. At every cell in the design space a material can be present or not. The second factor is the location of the fixed support points or normals the design will reset on. The third factor is the location the load or force is applied at. The fourth factor is the material cost. This penalizes the amount of material used in the design process and serves two purposes. The first is to generate non-trivial solutions. As we are maximizing the structure's stiffness the optimal solution is often to simply fill the entire design space with material. To generate more interesting designs, we must then limit the material through a cost penalty. The second is to reduce construction cost. A cost-effective structure will use the minimum amount of material to achieve a desired stiffness. Together these factors along with the choice in optimizer and random seed shape the design.

In practice it is common to work with continuous values for material density, $d \in [0, 1]$, rather than binary variables for material presence or absence at each element. This has the advantage of allowing for gradient based optimization methods. Denser elements can be viewed as "more" present and offer a greater contribution to the structures stiffness. After optimization, when looking to assemble the design elements with densities close to zero will be treated as voids (no material present) and values close to one will be filled with material. Thus during optimization it is key to push density values towards 0 or 1. The Solid Isotropic Material with Penalization method (SIMP) [15, 16] is a widely used approach to solve structural optimization problems. This approach operates on continuous densities $d \in [0, 1]$ and contains a penalty term $p = 3$ to steer elements towards 0 or 1. In this work we use the modified SIMP equation 1 from [17].

$$E_e(d_e) = E_{min} + d_e^p(E_0 - E_{min}) \tag{1}$$

Where $E_e$ is the stiffness coefficient or Young's modulus of the material at each element. $E_{min}$ is the minimum allowed stiffness of any element and will indicate voids in the final design. $E_0$ is the Young's modulus of a solid material. The goal in optimization is to maximize the stiffness of a design subject to a material cost and the application of a force. The parameters are the densities (presence of material) at each element. To evaluate the performance of the design we compute the design's elastic potential energy or compliance $c$, given in equation 2.

$$c(d) = U^T K U = \sum_{e=1}^{n} E_e(d_e) u_e^T k_0 u_e \tag{2}$$

Where $U$ is the displacement vector, $K$ is the global stiffness matrix and $KU = F$ where $F$ is the vector of applied forces. Compliance measures the structures displacement or deformation from the

force. A stiffer design will have a smaller compliance. In optimization we minimize the compliance to generate a stiffer structure. To impose a material cost $m$ we take a simple approach and add an additional term to the loss given by the MAE between the design's density $d$ and some specified target density $d^*$. This approach was chosen to allow independent weighting of the visual, physical, and material cost components of the total loss equation: 5.

$$m = |d^* - d| \tag{3}$$

The optimality and functionality of the generated design are important considerations. "Topology optimization problems have extremely many local minima"[18]. This can make it challenging to arrive at the globally optimal material layout for a given design problem. In the current work our focus is on generating functional designs rather than optimal designs. A functional design is a design that resists deformation and can be specified by a variable performance target for example a compliance threshold. For a given threshold there can be multiple possible functional designs. Generating and exploring the space of functional designs has previously been explored in the context of generating diverse competitive designs [19] and architectural design [20].

## 2.2 Generating art with CLIP

In this section, we provide a brief introduction into generating art with CLIP. CLIP [11] is a model for connecting images and text. Its architecture consists of an image encoder and a text encoder. During training the encoders are respectively fed images and their corresponding textual description, and the weights are optimized to maximize the cosine similarity between the text and image embeddings. After training the weights can be frozen and CLIP can be used to measure text image alignment. For example, given an image of a dog and two textual descriptions ["a dog" and "a cat"] we would expect the similarity between the image embedding and text embedding for "a dog" to be higher than the similarity of the image embedding and text embedding for "a cat".

We can then think of CLIP as a visual judge. Given some image and text CLIP can be used to score the perceived visual similarity of the image to the provided text. If we now hold the text description constant and replace the image with a blank canvas parameterized by learnable weights, we have the setup to an image generation problem. By maximizing the cosine similarity and backpropagating through the frozen CLIP image encoder we can update the canvas to generate images with features specified in the textual description. This approach has been widely used to generate artistic images and designs. Deep Daze [21] generates images by updating the parameters of SIREN an implicit neural representation network[22]. CLIPDraw [23] generates images with a differentiable render and learning vector stroke arrangements. [24] generates images using an evolutionary search algorithm and a CLIP loss to determine the placements of shapes. CLIP has also been used to generate diverse 3D objects [25–27].

## 2.3 Physics-based constraints for visual/physical design

Recent efforts have explored physics-based constraints for visual/physics design problems. Several approaches have focused on generating designs that maintain stability under gravity. Atlas3D [28] refines designs generated by a diffusion model using a physics-based loss formulated for standability. DSO [29] assembles a dataset of 3D objects and their stability scores and fine-tunes a diffusion model on this dataset. [30] uses an input image to guide the visual geometry of 3D objects and physics constrained optimization for stability and soft object deformation. Our approach differs in that our focus is on a different design problem, structural optimization, where the forces applied can vary in intensity and placement throughout the design space. Additionally, our approach does not require starting from an existing image or design generated by a diffusion model. Other efforts have looked to data-driven methods to incorporate differentiable physics constraints. Phy3DGen [31] trains a neural network to approximate a solid mechanics finite element solver. DrivAerNet++ [32] introduces a large multimodal dataset for aerodynamic car design for surrogate modeling and data-driven design. Our approach does not require data as the physics is differentiable.

In parallel to this work [33] examined CLIP stylization for topology optimization. However, the approach taken in [33] utilizes RGB channels and connected component labeling, which can cause overfitting to the visual domain. Our approach differs in that we do not use RGB values and the focus is on generating a shared binary (material presence/absence, black or white pixel) distribution. Additionally, we introduce an efficient physics based approach for removing unattached material

rather than connected component labeling. In Appendix Figure 9, we provided a comparison with our approach for an art deco building. In the design generated by [33] the windows exist only in RGB space and are painted over solid material, whereas in the design generated by TIDES the windows are structural features where there is no material present.

## 3   Text informed design

In Section 2, we provided background into two design tasks: Structural optimization a physics grounded approach for generating designs that resist deformation from an applied force and Generating art with CLIP guidance an artistic approach for generating visually interesting output from text. We argue that there is a natural overlap between these two design tasks. Both operate on a grid of pixels/elements and can be optimized with a differentiable performance metric. It then may be possible to share a grid of pixels/elements across both tasks and jointly optimize the design to maximize both its visual and physical performance.

We pose that linking a design's physical performance and visual appearance may offer a number of benefits:

1. **Text to drive design diversity and navigate the space of possible designs.** In standard topology optimization diverse designs are generated through randomizing or diversifying initial conditions. The suitability of the initial conditions is based on the shape of the design space which may or may not be known apriori. Text guidance may allow for the use of prior knowledge of structurally strong shapes given as text e.g. an "arch" or "hexagon" to steer the design process and reduce the number of runs an optimizer needs to reach a fit solution. Further, text guidance may provide a means of intelligently exploring a reduced set of initial conditions in a design space instead of relying on randomness to drive design diversity, e.g. in Figure 4C and D, we plot competitively performing designs that use different support strategies specified by text.

2. **Complex feature specification**. In a standard topology optimization approach, it is not clear how one would include complex visual design criteria in the optimization e.g. generating a tower that is both structurally sound and "resembles a robot". A visual loss specified by text may provide a mechanism for complex feature specifications.

3. **Mechanism for applying location based attention.** The placement of material in the image/design space depends on the locations a user has specified for the support and the force application. From an artistic perspective this could be used to ground the placement of aesthetic features and create focal points in the image where material must be placed to resist physical deformation. E.g. given an RGBA image, the alpha channel could define a structural optimization problem and be jointly optimized with a visual loss across the RGB channels.

In the current work, our focus is on the first two. However, to bridge a design's physical and visual performance there are two challenges we need to address. The first is mapping between the visual and material domain. CLIP was trained on 3 channel 224 by 224 RGB images while structural optimization operates on 1 channel densities bound between 0 and 1 where the design space may take various shapes and sizes. To resolve the size and shape differences we resize the design with bilinear interpolation. To resolve the channel difference, we repeat density channel 3-fold to generate a grey scale image. The inclusion of the visual loss term introduces an additional issue in that visually appealing images often make use of artistic techniques, such as shading, where pixel values are spread between 0 and 1. This is not ideal for structural optimization where it is desired for the pixels to occupy values close to 0 or 1, corresponding to material presence or absence.

To address this challenge, **we introduce a Hill function-style sigmoid** equation 4 and apply it to the output of the design encoding to push the densities close to 0 or 1. In Appendix C.2 Figure 10, we include examples of designs generated using our Hill function and with the Hill function removed. The Hill function removed designs contain intermediate grey scale values while the Hill function designs utilize values close to 0 and 1 indicating the Hill function is successful in pushing the densities to 0 or 1. The Hill function style sigmoid employed is given in equation 4 where the

values $\alpha = 0.8$ and $n = 20$ were determined heuristically.

$$d = \frac{1}{1 + \frac{\alpha}{x^n + 0.1}} \quad (4)$$

Further while CLIP was primarily trained on RGB images the training data includes images that are binary (black and white) in nature e.g. (silhouettes, outlines, shadow art, line art, and stencils). We append such terms to the end of the text prompts.

The second challenge is to ensure that all pixels contribute structurally to the design. The introduction of a visual loss term can lead to the creation of aesthetic features in the design that do not offer structural support. These features are often unattached to the main structure and float in space e.g. the "clouds" in Figure 1C. To solve this issue, **we developed a compliance based masking approach** to trim non-structural features from the design prior to showing it to CLIP. The compliance mask is generated by thresholding the element wise compliance returned from the physics simulator i.e. mask = log(compliance) $\geq$ threshold; threshold = -20. This returns a mask of zeros and ones where zero indicates a given pixel does not contribute structurally and one indicates it does. The mask is applied by multiplying by the design encoding. This ensures the CLIP loss only reflects pixels directly contributing to support and prevents over fitting to the visual loss through the creation of solely aesthetic features. In Appendix C.2 we generate designs with and without the compliance mask. The designs generated with the compliance mask contain no floating material while the designs generated with mask removed contain floating material indicating the masking approach ensures all pixels are connected to the design. In Figure 2, we sketch TIDES our text informed design framework. TIDES consists of three key components: a design encoding, a physics simulator, and a visual judge.

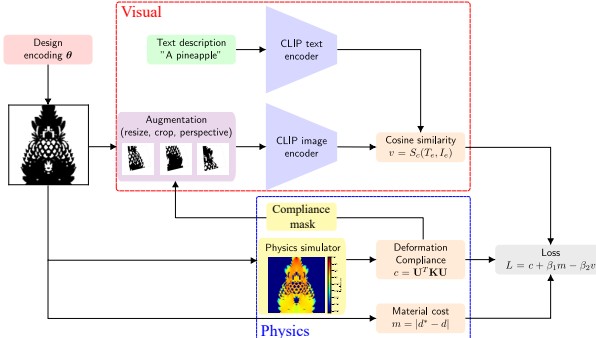

Figure 2: The proposed text informed design framework. The framework jointly optimizes a text to image visual loss, a physic-based loss, and a material cost.

The **design encoding** parameterizes the design space. Here, we modify a direct encoding approach commonly used in structural optimization. In this approach, there is one parameter for every element in the design space. At initialization, every parameter is assigned a value. The starting parameters can be a random canvas, zeros for an empty canvas, or ones for a solid canvas. We opt for a starting value of 1 as we can think of this as starting from a solid block of material. A Gaussian blur filter is then applied across the parameter grid. The filter is used to avoid checkerboard artifacts [34]. In a standard structural approach these densities would then be passed into the SIMP equation 1. Here we add the additional step of applying the Hill function-style sigmoid equation 4, to return the densities and push the densities towards 0 or 1.

The **visual judge** receives the output of the design encoding as input. A compliance mask is applied to zero out sections of the design that do not contribute structurally. The design is then repeated 3-fold channel wise to return a grey scale image. The image is then resized to 224 by 224, the dimension expected by the CLIP image encoder. The image augmentation scheme from [23] is then employed to generate a batch of images. The image is randomly cropped, random perspectives are taken, and the images are all resized. These operations are performed using the TorchVision library [35]. The batch of images is then passed through the CLIP Image Encoder, and the cosine similarity is computed between the image embedding and the text embedding. Randomly sampling sections of the images has the benefit of allowing TIDES to focus on the visual aspect of different sections of the design between trials. This enables TIDES to generate different designs when given the same prompt and problem setup in repeated trials.

The **physics simulator** receives the output of the design encoding as input and executes the modified SIMP approach, equations 1 and 2 to compute the compliance. We use the AutoGrad [36] SIMP implementation from [8] with an added wrapper to pass the gradients from autograd to pytorch.

The **material cost** is computed as the mean absolute error between the target density and the current density. Intuitively, we can think of the target density as the percentage of the design space the design should occupy if all pixels are either 0 or 1.

The **loss function** for TIDES is given in equation 5. Where $m$ is the material cost, $c$ is the compliance, $v$ is the vision loss given by clip score and $\beta_1$ and $\beta_2$ are heuristically determined values for weighting the material cost and visual loss.

$$\mathcal{L} = c + \beta_1 m - \beta_2 v \tag{5}$$

## 4 Experiments

In this section, we present experimental results for designs generated with TIDES. We include high-resolution plots of all designs in Appendix C along with detailed descriptions of the methods and experimental setup in Appendix B. The results cover structural design problem classes including single point force application, multiple point force application, multiple points and varied force application, and for 3D printing the 3-point bending problem used to test for beam strength. Additionally, in appendix B.6, we include results for starting from and tuning an existing design to improve physical performance.

### 4.1 Designs generated at different resolutions

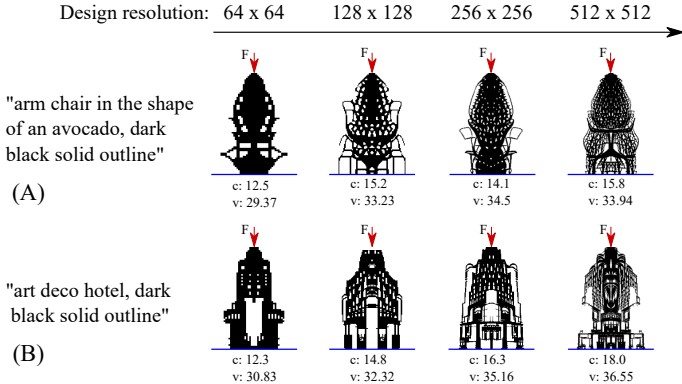

Figure 3: Generating designs at increasing resolutions for a tower structural optimization problem. A force is applied at the center of the design space indicated by the red arrow and the design rests on support given by the blue line. As the resolution increases feature complexity increases.

The pretrained visual judge (CLIP) was trained on 3 channel fixed resolution $224 \times 224$ images, while the design space in structural optimization operates on 1 channel and may take on various shapes and sizes. To evaluate TIDES ability to map between the visual and material domains and scale to structural design problems below or above the fixed resolution input of CLIP, designs were generated for a tower structural optimization problem at increasing resolutions from $32 \times 32$ to $512 \times 512$.

In Figure 3A, we plot designs generated from the text prompt: "arm chair in the shape of an avocado, dark black solid outline". Here, TIDES returns designs that increase in feature complexity with resolution. At $64 \times 64$ the returned design is a simple silhouette outline of a chair with a bulbous shape, at $128 \times 128$ texture emerges in the form of small cut outs and by $512 \times 512$ the design captures the pitted skin surface of an avocado. The designs are composed of two regions; the round bulbous shape occupies the top half of the design and the stand-like structure on which it rests at the bottom. The compliance of the TIDES designs are orders of magnitudes lower than the vision alone results Appendix C.3 Figure 11A and are within the same magnitude of the physics alone results Appendix C.3 Figure 11C, indicating the designs are physically sound while utilizing complex structural support features supplied by text.

In Figure 3B, we plot designs from the text prompt "art deco hotel, dark black solid outline". Here, feature complexity again increases with resolution. At $64 \times 64$ the design returned is a simple rectangular shape with square window-like cutouts, at $128 \times 128$ the detail increases and the design resembles the triangular shape of the famous Flatiron building in New York and by $512 \times 512$ the tower rises above a detailed marquee. The designs share the perspective of an individual looking up at the tower. The compliance of the TIDES designs are orders of magnitudes lower than a vision alone result Appendix C.3 Figure 11B and are within the same magnitude of the physics alone results Appendix C.3 Figure 11C while utilize complex support features supplied by text. This suggests that the mapping approach and Hill function design encoding employed in TIDES allow for joint optimization of a design's visual and physical properties across different scales.

## 4.2 Navigating the design space with text

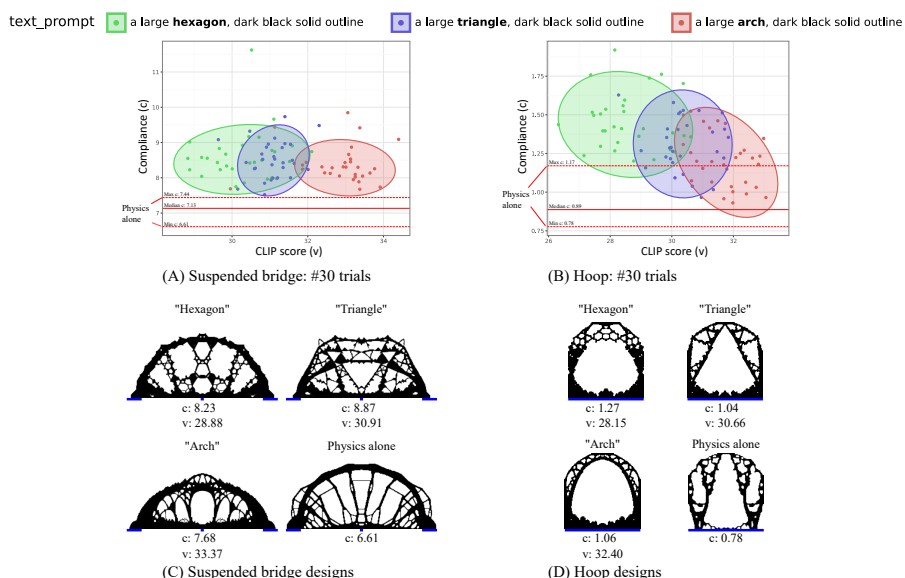

Figure 4: Distribution of design performance across 30 trials. (A) and (B) results for a suspended bridge and hoop design problem, the 90% confidence ellipse is plotted for each trial. (C) and (D) examples of designs returned. All 30 designs for each trial can be found in the Appendix C.5 and C.6.

For a given design problem, there are multiple possible paths an optimizer can take to achieve a functional design. To assess text as a means of navigating the design space and generating diverse designs, we generated designs for a suspended bridge and hoop design problem, Appendix B.2 & B.3. 30 trials of the TIDES approach were conducted for three different text prompts selected to elicit structural support features: "a large triangle, dark black outline", "a large hexagon, dark black outline", and "a large arch, dark black outline". Additionally, 30 trials were conducted for a physics loss alone, where the vision component was removed and the starting canvas was randomized to generate different designs between trials.

In Figure 4A & B, we plot the resulting design distributions. We plot a design for each text prompt trial in 4C & D, a plot of all designs can be found in Appendix C.5 & C.6. The designs returned for the suspended bridge and hoop design problems visually display and structurally utilize the support strategies specified by text, e.g. in Figure 4C & D: the designs generated from the "Hexagon" prompt utilize a hexagonal mesh, the designs generated from the "Triangle" prompt utilize a large central triangle and surrounding smaller triangles, and the designs for the "Arch" prompt utilize a large central arch and surrounding smaller arches. For the suspended bridge design problem we observe similar physical performance across designs utilizing different support strategies and features specified by text, with designs generated by TIDES approaching the upper bound of the physics alone results in terms of compliance suggesting that designs resist deformation. For the hoop design problem, the designs returned for the arch and triangle prompt overlapped in performance with the physics alone results. This is a point of interest, as it is expected that inclusion of a visual constraint

into the structural design problem will lead to some trade off in physical performance and suggests in some cases the addition of a visual text based loss may act as a mechanism for dislodging designs stuck at a local minimum.

## 4.3 Diverse design problems and text prompts

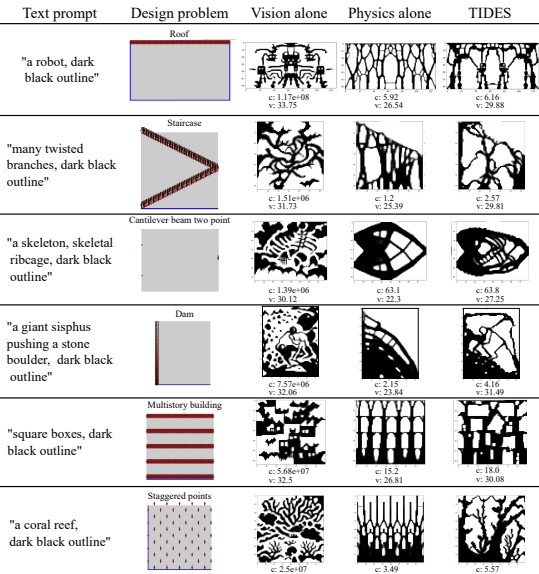

Figure 5: Generating designs for diverse initial conditions (force and support positions and text prompts). The designs resist deformation from the applied force while displaying features specified by text suggesting TIDES is robust to changes in the initial conditions. In Appendix: B.5 we provide the configurations for each design problem.

As input, TIDES receives a text prompt, target density, and the locations of the supports and applied forces. In Figure 5, we plot designs generated for a series of different text prompts and support and force positions. Across all structural design problems and text prompts: the vision alone generated designs display the requested visual feature but have no understanding of physics, the physics alone generated designs resist deformation and utilize simple support strategies that have no understanding of the visual design goal, and the TIDES generated designs are both physically sound, compliance values within the same order of magnitude of the physics alone results, and utilize complex support features specified by text. This indicates that TIDES is robust to changes in the initial conditions.

## 4.4 3D printed beams for a 3-point bending problem

The experimental performance of the TIDES approach was assessed by 3D printed beams subjected to three point loading. The 3D printing and testing setup is described in Appendix B.4. In Figure 6A, we plot the final designs generated by binarizing the TIDES and the physics alone output densities to 0 or 1. In Figure 6B we plot an example of the beams returned from the 3D printing process, 3 replicate beams were produced and tested for each design to account for variations in the printing process. In general, the rather fine features presented in this manuscript lead to some flaw development associated with terminations in paths of material when any dimension is not a discrete numeral of the path width (200 microns). At locations where this occurs void content will create stress concentrations and reduce part stiffness and strength. The x-direction strain field (left-right in the image) is presented in Figure 6D at a displacement of around 2.66 mm which is about the onset of non-linearity which can be seen in Figure 6C. The magnitude of the maximum measured strains is of the same order with the distribution and location of maximal values dependent on the morphology.

From the force-displacement graph, Figure 6C, the physical performance of the TIDES and Physics alone are the most similar in the 0 to 1.5 mm region of loading with some divergence from 1.5 to 2.66 mm. The physics alone design had the lowest compliance value of 1.0e-3 N/mm followed by

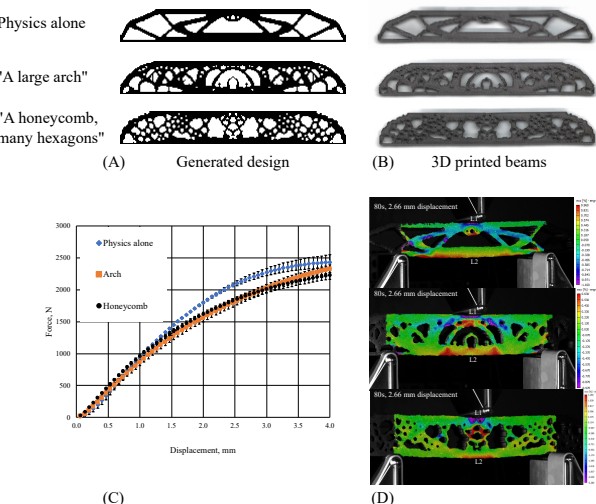

Figure 6: 3D printed beams tested on a 3 point bending problem. (A) Generated designs from TIDES and physics alone. (B) 3D printed beams returned from extruding the design along the y-axis. (C) Force-displacement behavior from the 3 point bend testing. The points correspond to the mean force and the bars indicate the minimum and maximum force value observed across the three replicates per trial. (D) Snapshot of beams under load. The beams rests on two separated points and a force is applied at the center.

the "honeycomb, many hexagons" with 1.13e-3 N/mm and the "large arch" 1.19e-3 N/mm. This aligns with the simulation results where the physics alone result returned the lowest compliance value of 280.0 c, note the simulation compliance values have a different unit than the experimental, in simulation the compliance is computed for every element and the summed value is reported and experimentally displacement is measured for the marked points in Figure 6D (L1-L2) and used to compute compliance. In simulation the "large arch" design returns a slightly lower compliance value of 285.71 c to the "honeycomb, many hexagons" design's 299.55 c this differs from the experimental results where the "honeycomb, many hexagons" design returns a slightly slower compliance than the "large arch" design. This could be due to manufacturing as previously mentioned. The performance across both the experimental and simulation for all trials are within the same order of magnitude and indicate that all designs successfully resist deformation from the applied force.

## 5    Discussion and Conclusion

In this work, we presented TIDES, an approach for bridging the design domains of physic agnostic text-to-image generation and physics constrained structural optimization. To the best of our knowledge, this is the first approach to jointly optimize a visual and physics-based loss for a load bearing design problem. Experimental results across a series of text prompts and structural optimization problems operating under different load and support conditions, at different resolutions, over repeated trials, and experimentally in the lab through 3D printing, show TIDES generates designs that resist physical deformation from an applied force while utilizing support strategies and features conveyed by text and is robust to changes in the initial conditions. This suggests that placing physical constraints on a pretrained text-image model is a valid new approach for ensuring that the generated design is physically sound and indicates that it is not necessary to train a new text to design model from scratch on physics data to generate sound designs. Further, from a structural design perspective, it suggests that embedding a visual constraint into structural optimization allows for text informed co-design and diverse design generation and is a valid mechanism for conveying complex design goals, such as using a specified structural shape, e.g."Hexagon", "Arch", and "Triangle" in the support strategy or a more aesthetic driven goal such as "in the shape of an avocado" or "art deco hotel", while respecting the underlying physical constraints. We believe that linking text-image models and differentiable physics simulators is an exciting new area of research towards generating physically sound designs that can be realized in life.

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

# Appendix

## A    Limitations and Impact Statement

This paper presents work whose goal is to advance the field of generative design by bridging the domains of physics constrained structural design and unconstrained text-image generation. As TIDES is a generative framework that uses CLIP, a pretrained text-image model to guide the designs it produces, there are the inherent risks of misuse and the amplification of biases present in the training data. Our approach has the added risk of allowing the generation of load bearing designs that could be constructed through means such as 3D printing. Our approach is currently limited by the simulation environment restricting 3D designs to layering of 2D force applications. Future work could extend the approach to simulators that support 3D meshes and higher resolution solvers and could have impacts on 3D design domains such as engineering, sculpting, and architecture.

## B    Design problem details and methods

### B.1    Tower design problem

In the tower design problem, Figure: 1, the design rests on the ground and resists deformation from a downward force applied at the top. A target density 0.3 is selected and the $x$ and $y$ dimensions of the design space are set at 128. As the force application is symmetric, we compute the compliance for half of the design and mirror the results in the image shown to CLIP. We used the text prompts: "Leaning tower of Pisa, dark black outline", "Eiffel tower, dark black outline", and "a stack of letters, dark black outline". In the loss function $\beta_1 = 50$ and $\beta_2 = 100$. The AdamW [37] optimizer was used with a learning rate of 0.25 and train for 100 epochs. Lastly we perform a quick manual pass to remove any disjoint artifacts created by the compliance masking, these disjoints are connected to the main structure, do not contribute structurally and occur where the mask occludes pixel with compliance over a specified threshold. In the varied resolution tower design problem, Figure: 3, the same setup is used and the design space is set to (64 x 64), (128 x 128), (256 x 256) and (512 x 512). The text prompts given to tides are as follows: "arm chair in the shape of an avocado, dark black solid outline" and "art deco hotel, dark black solid outline".

### B.2    Hoop design problem

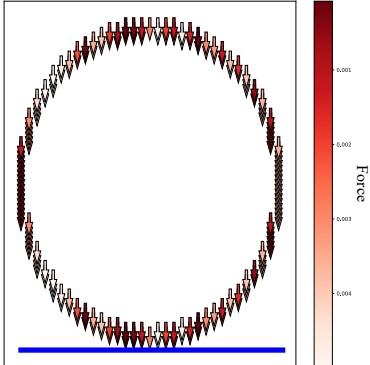

Figure 7: Hoop design problem setup

For the hoop design problem, Figure: 4B and 4D, the force and support layout is given in Figure: 7 the design rests on the ground and a variable a downward force is applied around the hoop. A target density 0.3 is selected and the the $x$ and $y$ dimensions of the design space are set at 128. As the force application is symmetric, we compute the compliance for half of the design and mirror the results in the image shown to CLIP. We use the text prompts: "a large triangle, dark black outline", "a large hexagon, dark black outline", and "a large triangle, dark black outline". In the loss function $\beta_1 = 50$

and $\beta_2 = 100$. The AdamW [37] optimizer was used with a learning rate of 0.25 and train for 100 epochs.

### B.3 Suspended bridge design problem

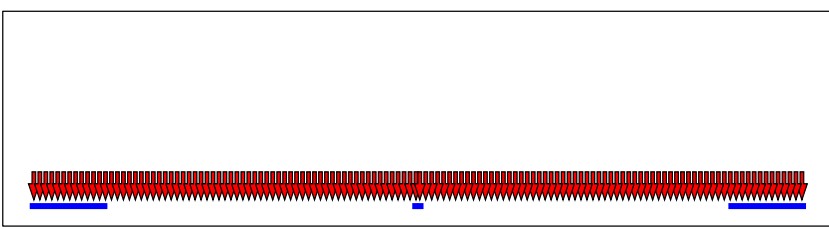

Figure 8: Suspended bridge design problem setup

For the suspended bridge design problem, Figure: 4A and C, the force and support layout is given in Figure: 8 the design rests on separated points and a downward force is applied along the span of the bridge. A target density 0.3 is selected and the dimensions of the design space are given as $x = 256$ and $y = 128$. As the force application is symmetric, we compute the compliance for half of the design and mirror the results in the image shown to CLIP. We use the text prompts: "a large triangle, dark black outline", "a large hexagon, dark black outline", and "a large triangle, dark black outline". In the loss function $\beta_1 = 50$ and $\beta_2 = 100$. The AdamW [37] optimizer was used with a learning rate of 0.25 and train for 100 epochs.

### B.4 3D printing and testing

For the beam design problem, Figure: 6A, the force and support layout is given in Figure: 6C the design rests on two separated points and a force is applied at the center. A target density 0.5 is selected and the dimensions of the design space are given as $x = 672$ and $y = 96$. This gives a (7 x 1) inch ratio in the 3D printed design. As the force application is symmetric, we compute the compliance for half of the design and mirror the results in the image shown to CLIP. We use the text prompts: "a large arch, dark black outline" and "a honeycomb, many hexagons, dark black outline". In the loss function $\beta_1 = 50$ and $\beta_2 = 100$. The AdamW [37] optimizer was used with a learning rate of 0.25 and train for 100 epochs.

Beams were printed using a Markforged X7 3D printer which extrudes nylon highly filled with short carbon fibers trademarked Onyx. Parts are build layer by layer starting with the paths of outer shell material infilling the center volume at alternating +45 and -45 deg. A path of material has a finite minimum width of 200 microns and a height of 100 microns limiting the minimum feature size. The beam rests on two lower rollers which move upwards in a displacement control test at 1 mm/min while the upper center roller is fixed. The center fixture has a 20 kN load cell that is used to measure the applied load. A more detailed description of the setup and methods are available in [38].

### B.5 Diverse design problems

For the roof design problem, the target density was 0.3 and the dimensions of the design space were $x = 128$ and $y = 64$. In the loss function $\beta_1 = 10$ and $\beta_2 = 250$. For the staircase design problem, the target density was 0.4 and the dimensions of the design space were $x = 64$ and $y = 64$. In the loss function $\beta_1 = 10$ and $\beta_2 = 250$. For the cantilever beam two design problem, the target density was 0.5 and the dimensions of the design space were $x = 80$ and $y = 64$. In the loss function $\beta_1 = 100$ and $\beta_2 = 250$. For the dam design problem, the target density was 0.5 and the dimensions of the design space were $x = 64$ and $y = 80$. In the loss function $\beta_1 = 10$ and $\beta_2 = 250$. For the multistory building design problem, the target density was 0.5 and the dimensions of the design space were $x = 70$ and $y = 64$. In the loss function $\beta_1 = 50$ and $\beta_2 = 250$. For the staggered point design problem, the target density was 0.5 and the dimensions of the design space were $x = 80$ and $y = 80$. In the loss function $\beta_1 = 50$ and $\beta_2 = 50$. The AdamW optimizer was used with a learning rate of 0.25 and train for 100 epochs for all additional design problems. The adjustments here to the weights on the visual loss and material cost for each trial were made to balance out an increase in the initial

physics loss in some of the more complex design problems. In B.1-B.4 the same weights ($\beta_1 = 50$ and $\beta_2 = 100$) are used for all trials.

## B.6 Starting from an existing design

Stable Diffusion 2.1 [39] was used to generate a 768 x 768 image for the text prompts: "the eiffel tower, dark black outline", "seattle space needle, dark black outline", "a robot, dark black outline", and "st louis arch, dark black outline". The default settings for Stable Diffusion 2.1 were used. The returned images were processed for compatibility with TIDES design encoding. The images were converted from three channel RGB to one channel grey scale and resized to 256 x 256 to match the design space of the structural optimization problem. This was then passed as the starting parameters for TIDES design encoding. A tower design problem B.1 was used. The target density and learning rate were adjusted to account for starting from an existing design. The learning rate was reduced and the target density was selected as a value less than the mean density of the starting image to account for the amount of unconnected material in the starting design that will be removed during optimization. For each text prompt the target density and learning raters are given as : "the eiffel tower, dark black outline" : learning rate = 0.02 and target density = 0.4, "seattle space needle, dark black outline": learning rate = 0.05 and target density = 0.14, "a robot, dark black outline": learning rate = 0.1 and target density = 0.4, and "st louis arch, dark black outline": learning rate = 0.02 and target density = 0.2. All other parameters used the same settings as B.1.

# C  Additional Results

## C.1  Comparison with [33]

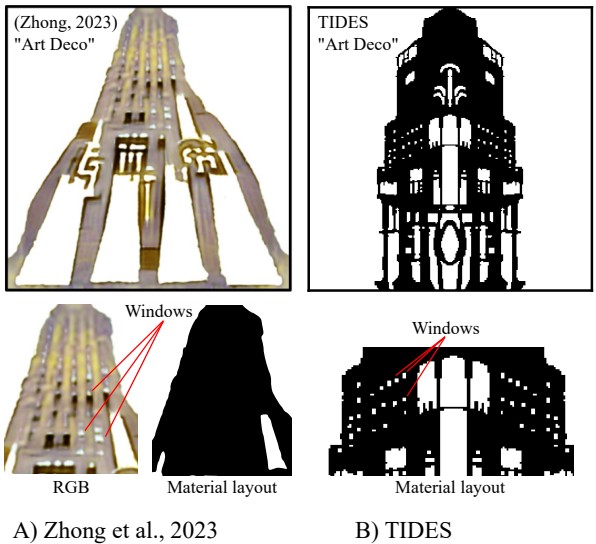

Figure 9: Comparison with [33] for an "Art Deco" structure. (A) [33] utilizes RGB channels and can overfit to the visual domain e.g. the generated Art Deco building has off-white colored windows. These windows are not structural features and are painted on over solid material. (B) TIDES does not use RGB values, the focus is on generating a shared binary (material presence/absence, black or white pixel) distribution. The window features present in TIDES all correspond to structural features (no material is present).

In [33] RGB channels are utilized during the design generation process. This can cause overfitting to the visual domain by allowing the model to improve the loss by painting over sections of the underlying structure rather than by shaping the physical structure itself. E.g. In Figure 9A, the generated art deco building has off-white colored windows. These windows are not structural features and are painted over solid material. Our work does not use RGB values, the focus is on generating a shared binary (material presence/absence, black or white pixel) distribution. The visual loss tends to

favor generating artistically pleasant images that utilize shading rather than binary values. To solve this, we introduced a Hill-function into the design encoding to control material distribution and push pixel/density values towards 0 or 1. E.g. In 9B, we present the design generated by our approach for an art deco building. In contrast to [33], the windows in our design all correspond to structural features (no material is present). [33] does not scale well to large design spaces and can result in designs with features that do not offer structural support. [33] utilizes the Connected Component Labeling (CCL) algorithm to remove unconnected features, this is computed at every optimization step. CCL scales poorly in computational time as the size of the design increases. In our work, we introduce a compliance based masking approach to remove unconnected features. As compliance is already computed for structural optimization there is no additional computational overhead required. Further compliance is physics based and allows the removal of features that are not offering direct structural support.

## C.2 Hill function and compliance masking ablation

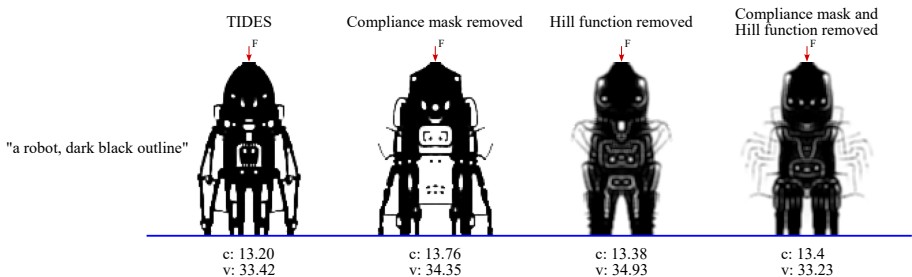

Figure 10: Hill function and compliance masking ablation. The compliance masking approach developed for TIDES ensures all features are connected. Removing this mask leads to designs with floating material. The Hill function developed for TIDES pushes density values to 0 or 1 (material presence or absence). Removing the Hill function leads to designs that utilize intermediate grey scale values between 0 and 1. See Figure 1 and Figure 11, for examples of designs generated from ablating the vision and physics loss components from TIDES.

## C.3 Designs generated at multiple resolutions

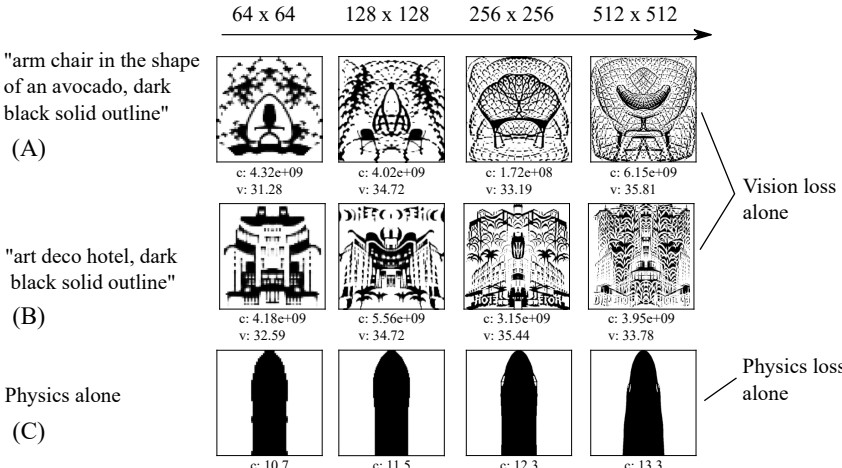

Figure 11: Loss ablation. Optimizing the tower structural optimization problem in Figure 3 for only a physic or vision loss. (A) and (B) are designs generated from maximizing the CLIP score. These designs have no concept of physics and return designs that have a compliance orders of magnitude higher then a sound design, and contain floating material. (C) The designs returned for minimize compliance. The designs returned are are relatively simple solid structures with a low compliance.

## C.4 Starting from an existing design

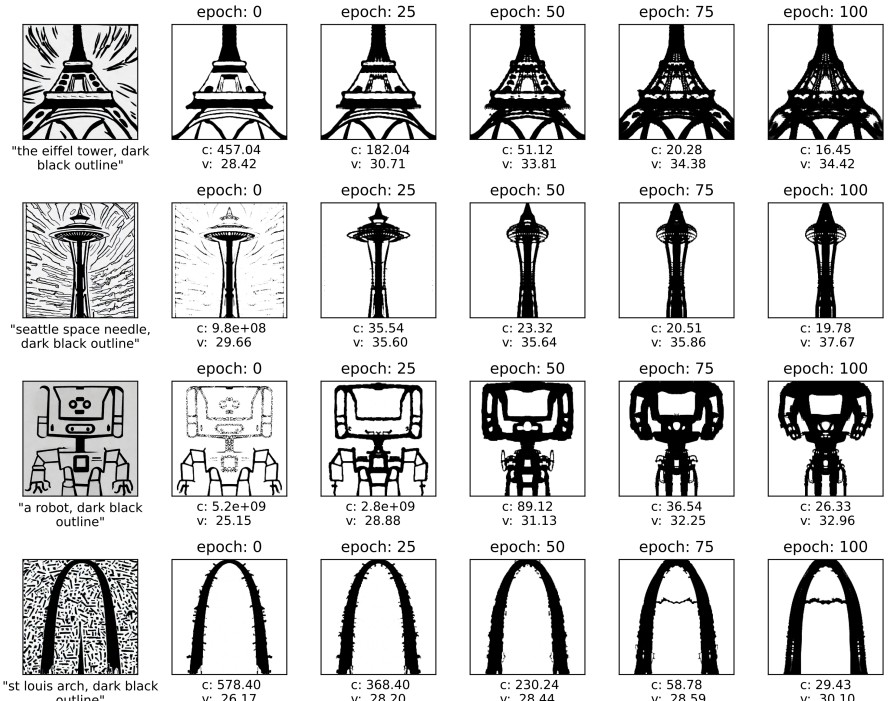

Figure 12: Starting from an existing design. An image generated by Stable Diffusion is converted to a single channel density representation and passed as the starting design encoding for a tower design problem. TIDES updates the material layout to resist the force applied at the top center of the design space while preserving the visual features given by text prompt and starting image.

In addition to generating a design from scratch, TIDES can be used to update an existing design to satisfy physical constraints. In Figure 12, we present the generation process starting from an existing design. Here, Stable Diffusion [39] is first used to generate an image from a text prompt. This image is then converted to a one channel grey scale image, resized to match the design space, and passed as the starting parameter for TIDES's design encoding. The "eiffel tower" and "st louis arch" starting designs contain floating material and span the entire design space. Over 100 epochs TIDES places material to improve compliance while maintaining visual features conveyed in the text prompt and starting design. The "eiffel tower" design maintains its iconic shape and the thickness of the support struts are increased by TIDES improving resistance to the applied force. The "st louis arch" design maintains the arch shape with an additional horizontal strut emerging during generation.

The "space needle" and "robot" starting designs represent a more challenging initial condition where, in addition to floating material, the designs are not fully connected and do not span the entire design space. The saucer in the "space needle" is floating above the tower base and the "robot" does not reach the top of the design space. In both cases, the starting designs do not resist the force applied at the top center and return very high compliance values. Over successive epochs, TIDES improves the design by removing floating material, connecting disconnected structures, and placing material to fully cover the design space and resist the applied force. The "space needle" preserves the visually distinctive feature of a saucer/orb supported by a pronged tower base. TIDES improves compliance during generation by increasing the thickness of the supports and reducing the size of the saucer. The "robot" maintains the rectangular head resting on a support body. The arm features do not offer much support and are removed by TIDES during generation with material reallocated and added to thicken the head and body to improve resistance to the applied force. Across all designs, the compliance mask rapidly removes the floating material and the hill function pushes a binary material distribution removing the grey background.

## C.5 Suspended bridge designs for all 30 trials

In this section we include high resolution plots of every design returned from 30 trials of the suspended bridge design problem. Each design corresponds to a point plotted in Figure 4A. To better view the detailed fine features present in each design we recommend zooming in on the individual design.

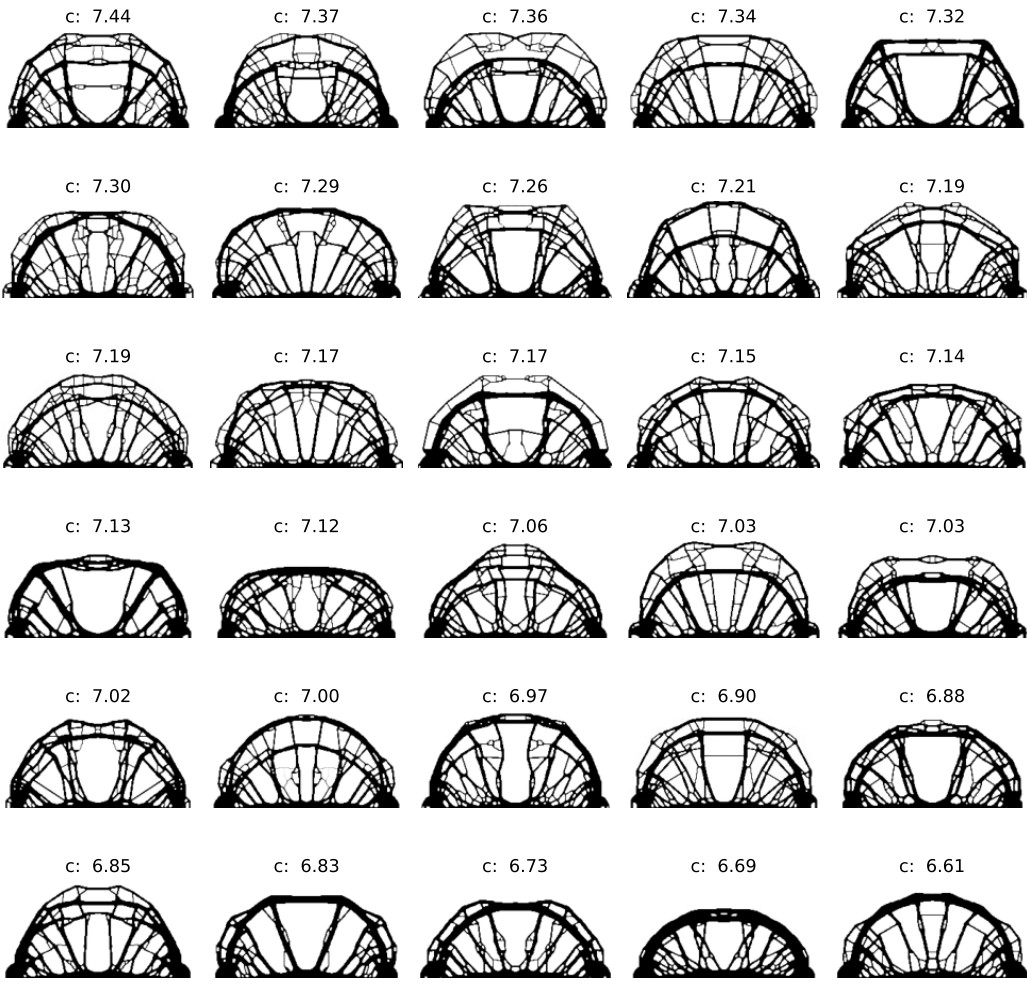

Figure 13: Designs generated from 30 runs of a physics loss alone for the suspended bridge design problem with a randomized starting canvas.

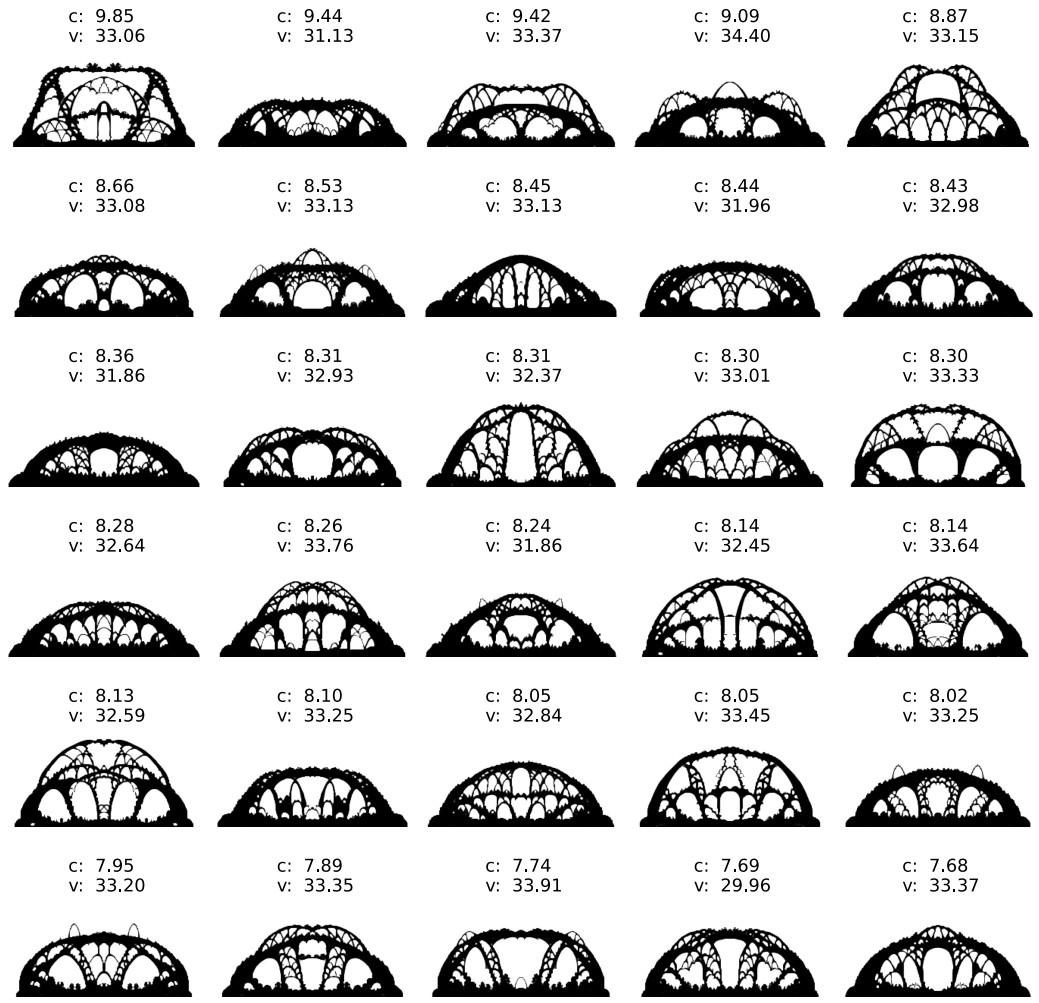

Figure 14: Designs generated from 30 runs of TIDES for the suspended bridge design problem given the text prompt "a large arch, dark black outline".

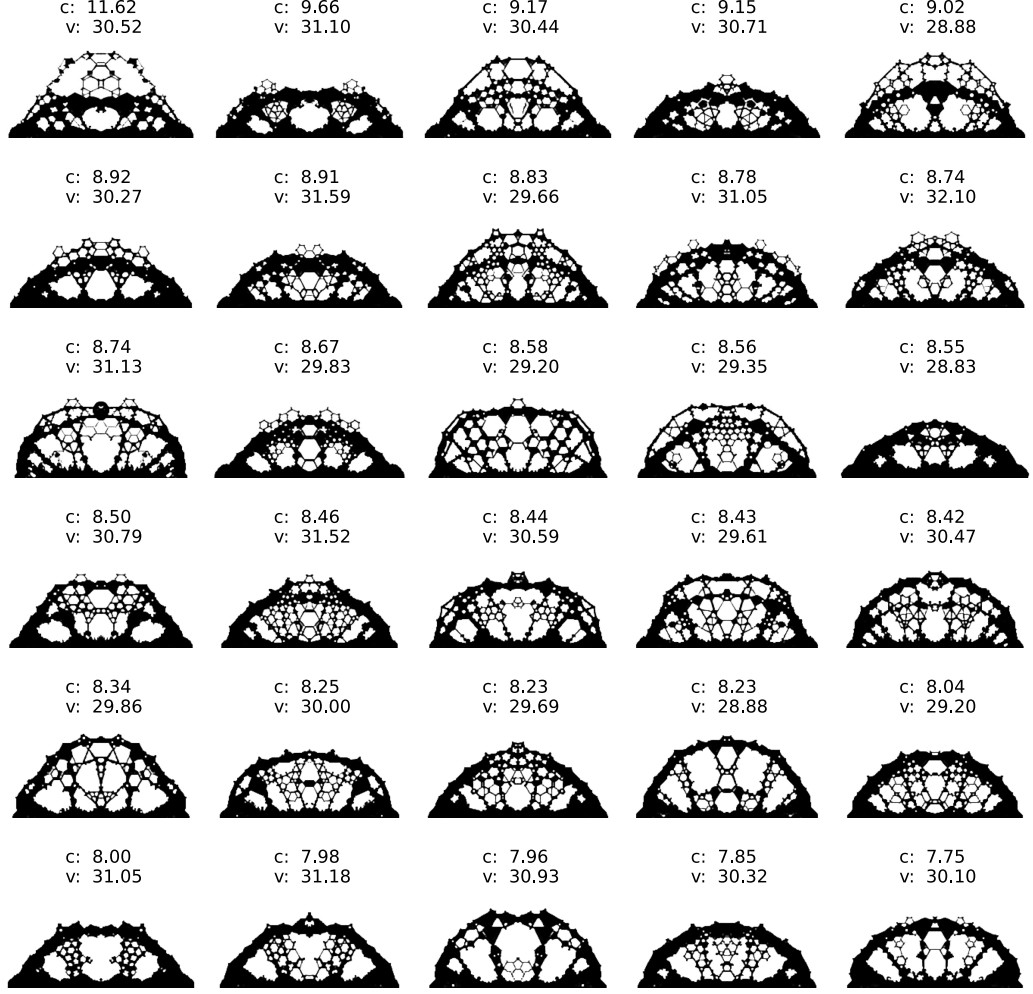

Figure 15: Designs generated from 30 runs of TIDES for the suspended bridge design problem given the text prompt "a large hexagon, dark black outline".

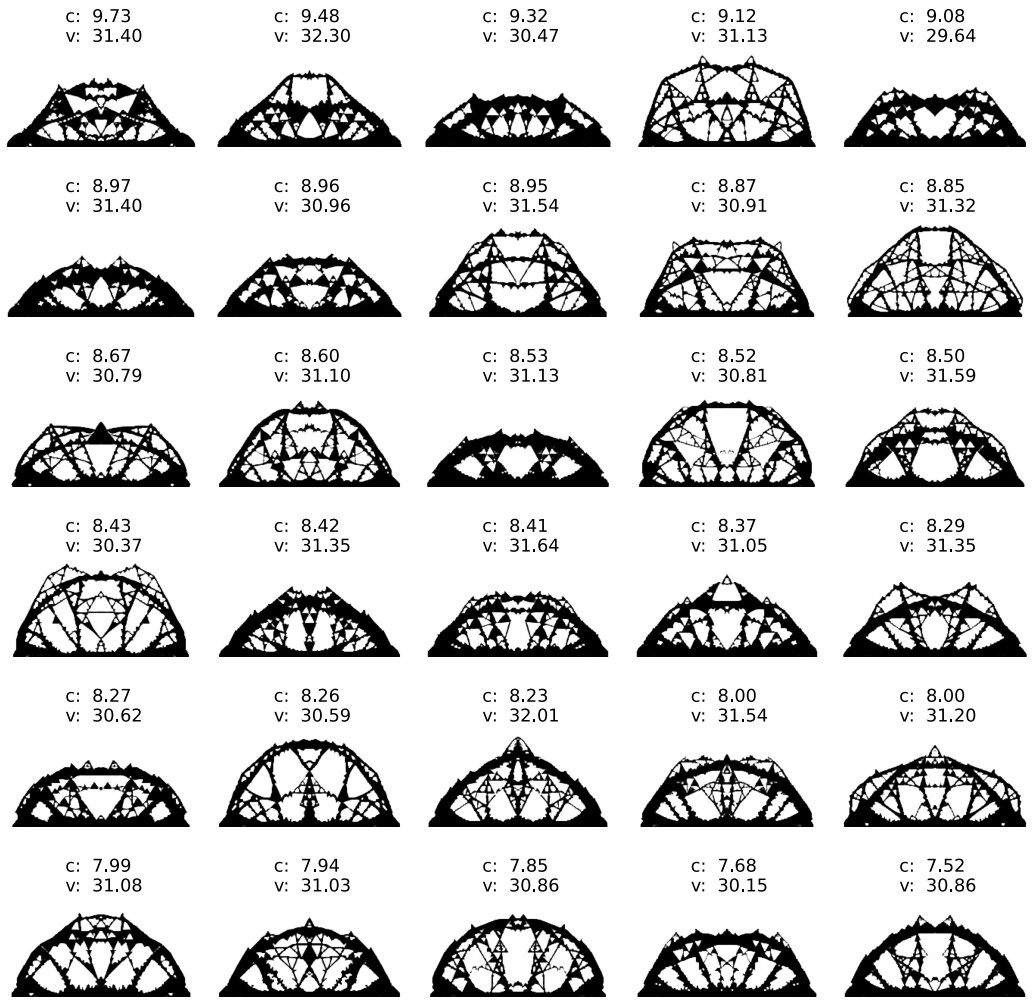

Figure 16: Designs generated from 30 runs of TIDES for the suspended bridge design problem given the text prompt "a large triangle, dark black outline".

## C.6   Hoop designs for all 30 trials

In this section we include high resolution plots of every design returned from 30 trials of the hoop design problem. Each design corresponds to a point plotted in Figure 4B. To better view the detailed fine features present in each design we recommend zooming in on the individual design.

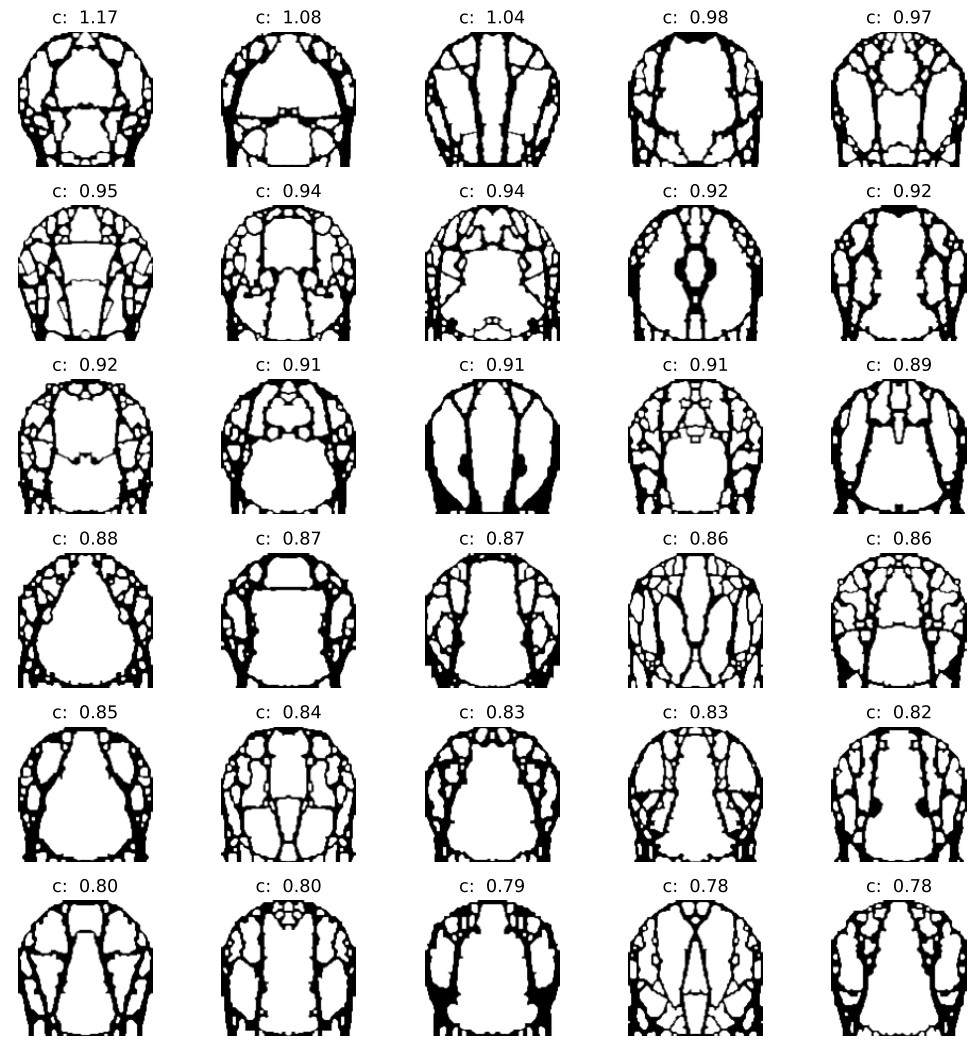

Figure 17: Designs generated from 30 runs of a physics loss alone for the hoop design problem with a randomized starting canvas.

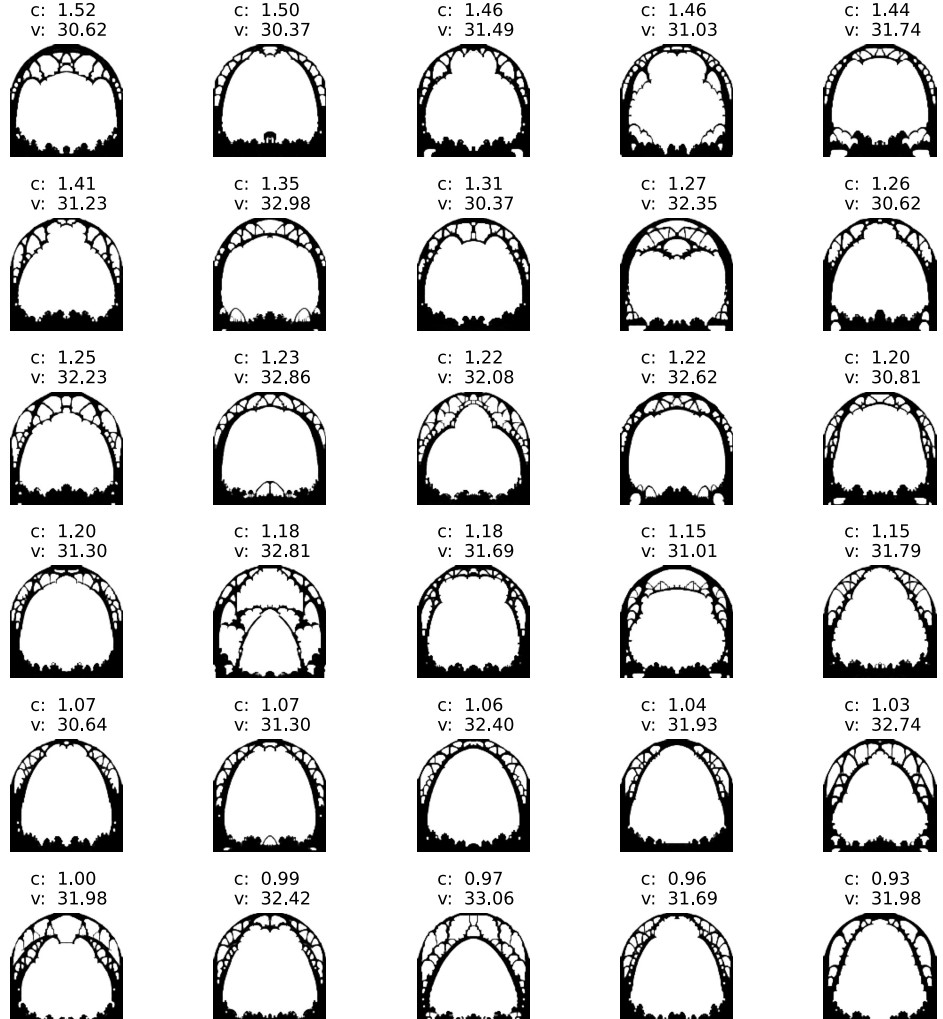

Figure 18: Designs generated from 30 runs of TIDES for the hoop design problem given the text prompt "a large arch, dark black outline".

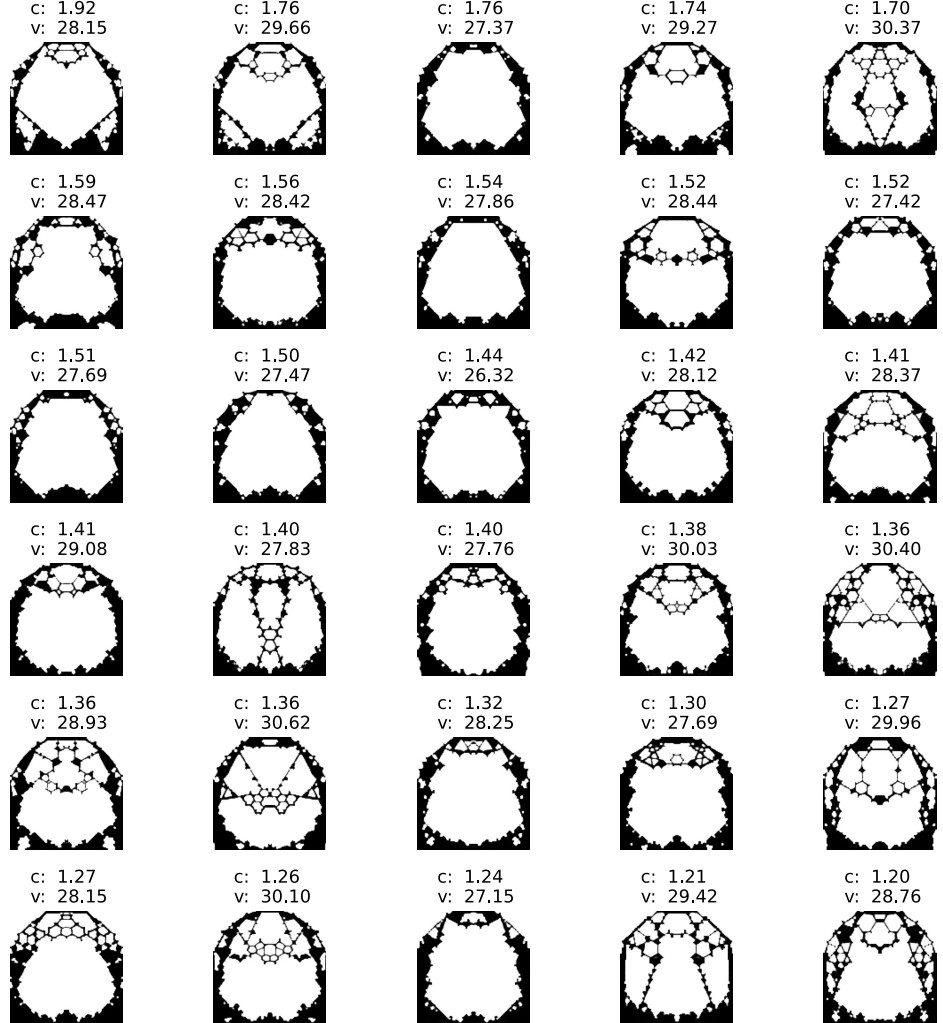

Figure 19: Designs generated from 30 runs of TIDES for the hoop design problem given the text prompt "a large hexagon, dark black outline".

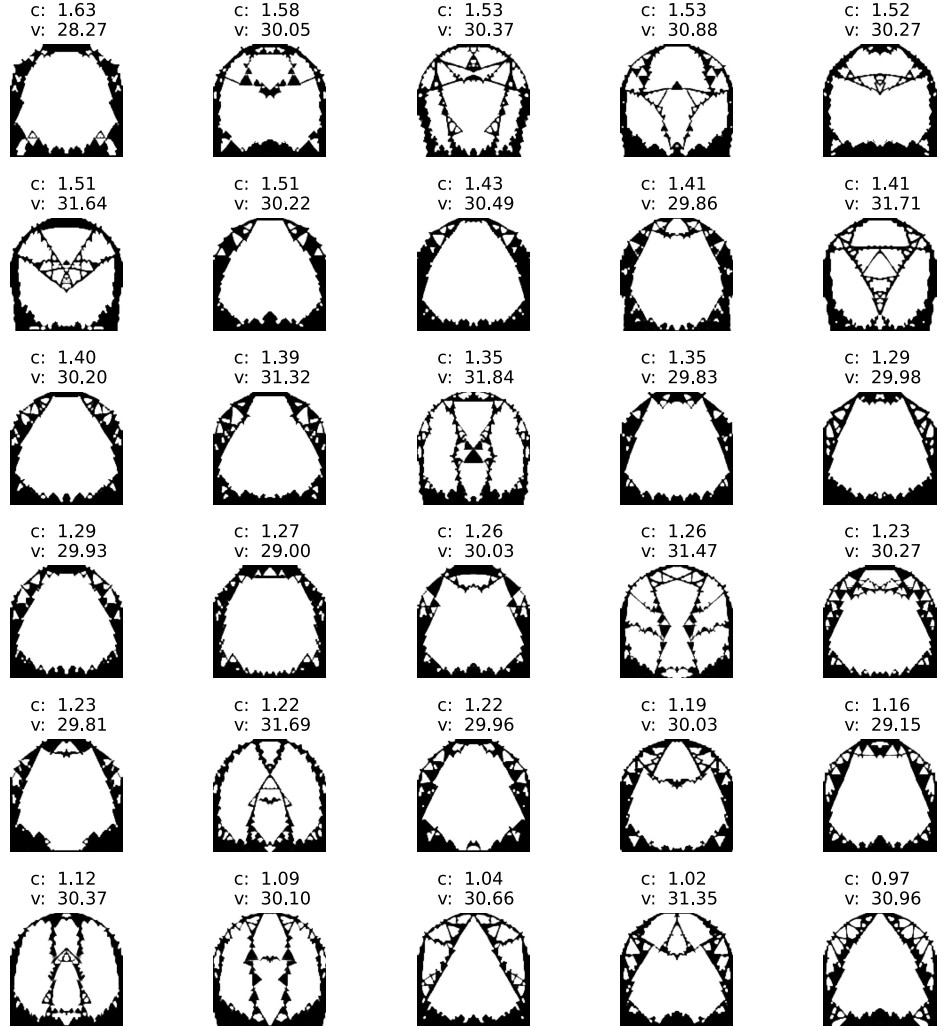

Figure 20: Designs generated from 30 runs of TIDES for the hoop design problem given the text prompt "a large triangle, dark black outline".

