# OpenReview forum: "Generating Physically Sound Designs from Text and a Set of Physical Constraints"
_NeurIPS.cc/2025/Conference — NeurIPS 2025 poster_

### Official Review · Reviewer_fywP · 2025-06-28

**Clarity:** 3
**Significance:** 3
**Originality:** 3
**Rating:** 4
**Confidence:** 3

**Summary:**

The paper proposed a pipeline for creating 2D designs that follows both the text description and physical constraints. It jointly optimizes a 2-D density grid with differentiable FEM simulation and a frozen CLIP similarity loss, so each design is both mechanically stiff and visually aligned with a text prompt. Experiments show it outperforms physics-only or vision-only baselines on the stiffness–CLIP-score trade-off and remains robust when 3D-printed and mechanically tested.

**Questions:**

1. Show a minimal 3D example—e.g., plugging your differentiable FEM module into a DreamFusion-style pipeline and optimizing a 3D structure?  This would directly demonstrate that the method generalises beyond 2D.
2. SDS-Loss Comparison: Replace the CLIP loss with the SDS loss and see if it provides better results.
3. Provide analysis on hyperparameter sensitivity. How hard is it for a user to find suitable parameters? This will reveal how robust the framework is and offer guidance to practitioners.
4. Timing and Scaling: Report the wall-clock time per iteration and total training time for each resolution (64 → 512), and compare with other solutions on the same hardware.

**Ethical Concerns:**

["NO or VERY MINOR ethics concerns only"]

**Final Justification:**

Thank the authors for their rebuttal. My concern for Q3 is resolved, but the rest are not:

Q1, extruding a 2D design to 3D is not real 3D. Would love to see results with more spatial constraints.
Q2 is rather easy to implement. It would be better to be included in this paper's revision rather than as future work.
Q4, please include a more detailed timing for specific settings.

Therefore, I decided to keep my current rating.

**Limitations:**

yes.

**Paper Formatting Concerns:**

No comments.

**Quality:**

3

**Strengths And Weaknesses:**

# Strengths
1. First to insert a frozen CLIP semantic loss into differentiable topology optimization, enabling joint vision-physics gradients.
2. Well-designed experiments with strong physics-only / vision-only baselines and 3D-printed real-world mechanical validation.
3. Offers a text-driven yet mechanically sound design pipeline, potentially impactful for architecture and mechanical engineering.
4. Pipeline diagram and loss equations are concise and easy to follow.

# Weaknesses
1. Limited to 2D structures; no convincing argument or demo for 3D extension.
2. Many tunable hyperparameters, but no systematic sensitivity study.
3. Alternative Design: Diffusion-based losses (e.g., SDS) could replace CLIP loss
4. Runtime: The paper states only the hardware used but omits per-iteration or total wall-clock times, making efficiency hard to judge.

---

> ### Author Rebuttal · Authors · 2025-07-30
>
> Thank you for your review.
>
> **W2. Many tunable hyperparameters, but no systematic sensitivity study.**
>
> There are 2 tunable hyperparameters: $\beta_1$ and $\beta_2$, the weight on the material cost and visual loss. The selection of these values has a subjective component as the visual appearance is a design goal. Given the speed of the algorithm at low resolution and that the parameters scale across resolution (Figure 3 all designs are generated with the same parameters), a user can quickly iterate over designs at low resolution to find the parameters that match their desired appearance and physical performance and then run at the desired resolution.
>
>
> **Q1. Show a minimal 3D example—e.g., plugging your differentiable FEM module into a DreamFusion-style pipeline and optimizing a 3D structure? This would directly demonstrate that the method generalises beyond 2D.**
>
>
> The three point bending problem for beam design can be simplified as a 2D problem with extrusion as in section 4.3. Here, the beams are designed with our approach, 3D printed through extruding the design, and tested under load for a 3 point-bending problem. The results for the 3D beam align with the performance in simulation validating the approach. A current limitation of this approach as mention in the paper is the simulation environment restricts 3D designs to layering of 2D force applications.
>
> **Q2. SDS-Loss Comparison: Replace the CLIP loss with the SDS loss and see if it provides better results.**
>
> The work under review presents the concept of linking a differentiable physics simulator and text-image model and demonstrate that this is a promising option for placing physical constraints on a pretrained text-image model. The success of a visual loss in this regard is a finding of this work. In future directions for this work such as placing physical constraints on a diffusion model, SDS could be used.
>
> **Q3. Provide analysis on hyperparameter sensitivity. How hard is it for a user to find suitable parameters? This will reveal how robust the framework is and offer guidance to practitioners.**
>
> All results in the main body of the text use the same hyperparameters. Here the visual weight was set as 100 as it is fairly common in the literature to weight the clip score by 100. The hyparameters have a subjective component given the visual appearance of the design is a design target. Given the speed of the algorithm at low resolution and that the parameters scale across resolution (Figure 3 all designs are generated with the same parameters), a user can quickly iterate over designs at low resolution to find the parameters that match their desired appearance and physical performance and then run at the desired resolution.
>
> **Q4. Timing and Scaling: Report the wall-clock time per iteration and total training time for each resolution (64 → 512), and compare with other solutions on the same hardware.**
>
> The slow step is the physics simulation, the visual loss runs almost instantaneously on the GPU. TIDES is slightly slower then physic alone baseline as you need to run both and pass the gradients between the CPU and GPU. Generation times varied from less than a minute for low resolution designs and up to 15 minutes for 512 x 512, a more efficient differentiable physics simulator could greatly speed this up.

---

> > ### Comment · Reviewer_fywP · 2025-08-04
> >
> > Thank the authors for their rebuttal. My concern for Q3 is resolved, but the rest are not:
> >
> > Q1, extruding a 2D design to 3D is not real 3D. Would love to see results with more spatial constraints.
> > Q2 is rather easy to implement. It would be better to be included in this paper's revision rather than as future work.
> > Q4, please include a more detailed timing for specific settings.
> >
> > Therefore, I decided to keep my current rating.

---

> ### Author Response · Authors · 2025-08-04
>
> Thank you for your response and providing additional details on your remaining concerns. We provide the requested timings for Q4 and additional clarifications for Q1 and Q2 below.
>
> **Q1. extruding a 2D design to 3D is not real 3D.**
>
> We make no claims of real 3D and to avoid confusion around 3D designs in the current work we state in the abstract: "2D beam designs that are extruded and 3D printed.". The MBB beam/3-point-bending problem we discuss in our response to Q1 is a widely used benchmark for structural optimization and is commonly extruded and validated through 3D printing in the structural optimization literature.
>
> For recent examples of the 2D MBB beam extruded to 3D as a structural optimization benchmark see:
> - "Multiscale topology optimization and 3D printing of continuous carbon fiber reinforced composites lattice structure" (2025)
> - "A novel multi-thickness topology optimization method for balancing structural performance and manufacturability" (2025)
> - "Topology optimization for material extrusion-based additive manufacturing processes with weak bead bonding" (2023)
>
>
> **Would love to see results with more spatial constraints.**
>
> In Appendix figure 10, we present additional designs from diverse 2D spatial constraints in terms of both support and force application including horizontal force application, random force application, suspend designs that do not rest on the ground and more. In practice, our approach is not limited to the design problems presented, and a user can supply their own 2D spatial constraints i.e. force and support placement locations as the initial conditions.
>
> **Q2. is rather easy to implement. It would be better to be included in this paper's revision rather than as future work.**
>
> SDS-Loss adds additional computational overhead (loading a diffusion model requires a GPU with high VRAM) over a CLIP loss (which can be run with a minimal GPU or on a CPU). A goal of this work is to make it easily accessible. The current approach generates designs that perform well visually i.e. the text-based features are readily identifiable while remaining accessible and easy to replicate with standard consumer hardware.
>
> **Q4. please include a more detailed timing for specific settings.**
>
> We have included the generation times for running the same text prompt and design problem at different resolution (Figure 3) below:
>
> Design resolution        |  TIDES         |   Structural optimization (baseline) | Run time difference (i.e. visual loss computation + gradient passing)|
> | :---------------- | :------: | :------: | :----: |
> |(64 x 64)                 | 27.25 sec      |         11.31 sec                    |     15.94 sec |
> |(128 x 128)               | 44.77 sec      |         22.975 sec                   |     21.795 sec |
> |(256 x 256)               | 170.72 sec     |         140.65 sec               |     30.07 sec |
> |(512 x 512)               | 876.19 sec     | 812.82 sec            | 63.37 sec |
>
>  The slow step is the physics simulation and is shared between structural optimization (baseline) and TIDES. TIDES is slightly slower as it requires the additional visual loss computation and gradient passing.

---

### Official Review · Reviewer_UJvm · 2025-06-30

**Clarity:** 2
**Significance:** 2
**Originality:** 2
**Rating:** 4
**Confidence:** 3

**Summary:**

This paper presents a generative framework that can synthesize physically sound structural designs according to text instructions. Given a design instance, it leverages a physics simulator to compute deformation compliance and material cost, and uses CLIP to obtain the similarity between the visual appearance and the text guidance. Then, it jointly minimizes a compliance loss, a material cost, and an image-text alignment loss to achieve a functional design that can withstand applied forces. Experiments on several design problems, as well as a 3D-printing evaluation, demonstrate the effectiveness of the proposed method.

**Questions:**

1.	The use of differentiable physics to enhance the physical soundness of generated assets has been explored in several existing methods [a-c]. What distinguishes the proposed method from these works?

2.	The paper starts with a design encoding value of 1. Is it possible to initialize the proposed method using existing designs (e.g., from artists) that may not initially withstand the applied force, and then optimize these designs with minimal modifications to ensure they can support the force? This can improve the practical usage of the method.

3.	The paper employs varying values for $\beta_1$ and $\beta_2$ across different design problems (see Section B.5), suggesting that the proposed method may be sensitive to the selection of hyperparameters. How are these hyperparameters chosen? Did the authors conduct experiments to determine a set of hyperparameters that perform well across all the considered problems?

4.	The experiments only compare the proposed method with its own ablations (w/o visual loss or w/o physics loss). How does the method compare with state-of-the-art text-to-image models like Stable Diffusion[d]?

5.	What is the running time for generating a single design?


References

[a] Physically Compatible 3D Object Modeling from a Single Image. NeurIPS 2024.

[b] Atlas3D: Physically Constrained Self-Supporting Text-to-3D for Simulation and Fabrication. NeurIPS 2024.

[c] DSO: Aligning 3D Generators with Simulation Feedback for Physical Soundness. arXiv 2025.

[d] High-Resolution Image Synthesis with Latent Diffusion Models. CVPR 2022.

**Ethical Concerns:**

["NO or VERY MINOR ethics concerns only"]

**Final Justification:**

The authors' response addresses most of my initial concerns, particularly regarding the distinction from existing work. The one remaining is the experiment using existing designs. Based on the authors' response, I decided to adjust my recommendation to 4.

**Limitations:**

Yes

**Paper Formatting Concerns:**

No paper formatting concerns.

**Quality:**

2

**Strengths And Weaknesses:**

**Strengths**

This paper presents an end-to-end framework for generating images of structures from text while ensuring physical accuracy. It leverages the strengths of a multi-modal language model, CLIP, to enhance the alignment between text and images. Additionally, it incorporates a differentiable physics simulator to optimize the structural design for physical soundness. The overall concept is straightforward and effective, as demonstrated by real-world 3D printing experiments.

**Weaknesses**

There is ample room for improvement in the paper's structure and organization. Specifically, it lacks clear sections and many parts are mixed together, making it difficult for readers to understand. For example, in Section 2, background and related work are intermixed, making it difficult to find an introduction to similar existing works on the same task. Similar issues can also be found for Section 3, and it is suggested to use subsections to organize the paper better.

Additionally, there are concerns regarding the technical novelty and completeness of the evaluation of the proposed method; see the questions below for details.

---

> ### Author Rebuttal · Authors · 2025-07-30
>
> Thank you for your review.
>
> **W1 For example, in Section 2, background and related work are intermixed, making it difficult to find an introduction to similar existing works on the same task.**
>
> In Section 2 we provide an overview of two independent design problems structural optimization (2.1 similar existing work from the physics side) and  text-image generation with CLIP (2.2 similar existing work for the text-image side).  In Section 3 we make the case there is value in combing these two design problems into a new task. The field is rapidly progressing and at the time of writing there were no existing works on embedding physical constraints into text-image models. The reviewer has brought to our attention very recent concurrent works [a-c] which use a physic loss to tackle a different design problem (stability and image guided design). We have updated section 2 to added a third subsection titled "physics constrained generative design" to our related work section based on our response to question 1.
>
> **Q1. The use of differentiable physics to enhance the physical soundness of generated assets has been explored in several existing methods [a-c]. What distinguishes the proposed method from these works?**
>
> The first version of the manuscript under review predates [a-c]. The current submission has been updated to address several of the challenges of linking CLIP with a finite element solver. Below we discuss how our work differs from [a-c].
>
> [a] Requires an input image to guide the visual geometry of the physical object returned. Our work differs in that it does not require an input image, we start from a solid block of material and use  a differentiable physics simulator, material cost, and visual loss given by a text prompt to shape the design's structure during optimization. In addition our work is also generative, in that in repeated trials for the same physical constraints and text prompt, our approach generates a distribution of diverse designs e.g. Appendix C.4, C.5 that utilize the text informed feature to satisfy the physical constraint through various support strategies rather than adhering to a specific predefined visual geometry.
>
> [b] and [c] are concerned with generating stable designs, designs that remain upright. In [b] a physics-based loss is formulated for standability and used as regularizer to refine designs for stability. In [c] a dataset of 3D objects with stability scores is generated and used to fine tune a generator. The designs in [b] and [c] rest on a continuous flat or curved surfaces.
>
> Our work differs from these methods in that our focus is on a different design problem, structural optimization. In structural optimization, the goal is to generate a design that resists deformation from an applied force through minimizing compliance. The forces applied in structural optimization can vary in intensity and placement throughout the design space rather than a constant gravitational force. For example, in Appendix Figure 10 we present designs generated from a variety of force applications including: horizontal force application (Dam), multiple point force application (multistory building), varied force application (staircase). Further, the placement of supports is not limited to continuous flat or curved surface e.g. the suspension bridge (Figure 4) or the cantilever beam two-point design (Appendix Figure 10), are suspended and sections of the design do not rest on the ground.
>
> **Q2. The paper starts with a design encoding value of 1. Is it possible to initialize the proposed method using existing designs (e.g., from artists) that may not initially withstand the applied force, and then optimize these designs with minimal modifications to ensure they can support the force? This can improve the practical usage of the method.**
>
> Yes, as the design encoding is a direct encoding an existing design could be used as the starting point for optimization. To use an existing designs, all one would have to do is resize the existing design to the same size as the desired design space and pass this as the initial state for the design encoding. We opted to use a starting value of 1 as it offers the nice intuitive interpretation of starting from a solid block of material. For best performance with an existing design we would recommend computing the ratio of black/white pixels in the existing design and setting the material cost higher than this value.
>
> **Q3. The paper employs varying values for and across different design problems (see Section B.5), suggesting that the proposed method may be sensitive to the selection of hyperparameters. How are these hyperparameters chosen? Did the authors conduct experiments to determine a set of hyperparameters that perform well across all the considered problems?**
>
> All results in the main body of the paper use the same parameters (B.1-B.4). The visual loss weight of 100 was selected as it is fairly common in the literature to weight the clip score by 100. In Appendix Figure 10 we increased the visual loss and adjusted the material cost to balance out an increase in the initial physics loss in some of the more complex design problems. Here there is still overlap in parameters between design problems, 5 of the 6 problems use the same visual weight, 3 use the same material weight of and another 2 use the same weight of 50 the same value used for (B.1-B.4). The method is not particularly sensitive to hyperparameter selection and given the design algorithm is fairly quick and the subjective nature of the visual design goal we found in practice a user can quickly iterate over a couple of parameters and select the one they prefer. We have updated section B.5 to clarify the parameter selection.
>
>
> **Q4. The experiments only compare the proposed method with its own ablations (w/o visual loss or w/o physics loss). How does the method compare with state-of-the-art text-to-image models like Stable Diffusion[d]?**
>
> State-of-the-art text-to-image models like Stable Diffusion cannot be directly compared with the proposed method as they operate over the visual domain (3 channel RGB images), do not utilize density constraints or physical constraints and are unable to generate physically meaningful designs. The w/o physics loss designs we compare against can be evaluated with the physics simulator as they still operate over 1-channel, satisfy the density constraint, and are pushed by our encoding to a binary pixel distribution.
>
> **Q5. What is the running time for generating a single design**
>
> The slow computational step is the physics simulator on the CPU. The visual loss runs almost instantaneously on the GPU. Generation times varied from less than a minute for low resolution designs and up to ~15 minutes for 512 x 512, a more efficient differentiable physics simulator could greatly speed this up.

---

> > ### Comment · Reviewer_UJvm · 2025-08-05
> >
> > Thank you for the response, which addresses most of my initial concerns. Please make the background and related work more organized, and include the above discussion in the final revision.
> >
> > I also wonder if it is possible to supplement some experiments using existing designs as discussed in Q2? I think such experiments could further validate the practical usage of the method.

---

> ### Author Response · Authors · 2025-08-05
>
> Thank you for your response. We are glad to hear that we addressed your initial concerns. We will make the requested changes to the background and related work section. As we discuss in Q2 starting from an existing design is easy to implement and we will run this experiment and included it in the additional results section Appendix C.

---

### Official Review · Reviewer_nNGQ · 2025-07-01

**Clarity:** 3
**Significance:** 2
**Originality:** 2
**Rating:** 2
**Confidence:** 3

**Summary:**

The paper presents TIDES, a differentiable pipeline that jointly optimizes a CLIP-based visual loss with structural compliance and material-cost losses so that 2D density fields evolved from a prompt end up both visually faithful and mechanically stiff. Overall, the study demonstrates that adding differentiable physics to a pre-trained generative model can produce manufacturable, prompt-conditioned structures.

**Questions:**

Please refer to the "Weaknesses" section.

**Ethical Concerns:**

["NO or VERY MINOR ethics concerns only"]

**Final Justification:**

The rebuttal clarifies the method’s novelty and addresses several technical questions. However, important related works on physics-constrained generation are not sufficiently discussed, and comparisons are missing. Furthermore, evaluation on existing datasets such as DrivAerNet++ could better demonstrate generality, especially for 3D domains with stricter physical constraints. While the contributions are interesting, these gaps prevent a higher score. I will maintain my current score.

**Limitations:**

yes

**Quality:**

2

**Strengths And Weaknesses:**

**Strengths**:
* The paper explicitly co-optimize a differentiable finite-element compliance loss (modified SIMP) and a CLIP text-image similarity loss in a single end-to-end loop, which unifies topology optimisation and text-to-image generation.
* This article provides a detailed explanation of the hyperparameters and visualization results of the experiment, and it has strong reproducibility. Meanwhile, conduct a thorough analysis of the potential limitations and impacts.

**Weaknesses**:
* All 3D results are 2D densities simply extruded,  so claims of “physically sound designs” are somewhat overstated for real engineering practice.
* Comparisons are limited (limited comparison methods and datasets/tasks). The paper omits stronger co-design baselines such as recent diffusion-based inverse design methods, making it hard to judge relative performance gains.
* The manuscript’s novelty derives mainly from combining these known techniques in one pipeline (most core components—SIMP compliance optimization, CLIP-based text supervision and Hill-type density projection are adopted from earlier papers.), not from introducing new algorithms or theory, which is more of a incremental. It would be best to explain what the methodological contributions are.

---

> ### Author Rebuttal · Authors · 2025-07-30
>
> Thank you for your review.
>
> **W1. All 3D results are 2D densities simply extruded, so claims of “physically sound designs” are somewhat overstated for real engineering practice.**
>
> We make no claims regarding real engineering practice, the field is not there yet for text based design. Structural optimization is widely used to design physical structures and TIDES generated designs approach the upper performance bound of designs generated by structural optimization, further we take the additional step of printing physical versions of the designs and validating the physical performance under load in the lab.
>
> **W2. Comparisons are limited (limited comparison methods and datasets/tasks). The paper omits stronger co-design baselines such as recent diffusion-based inverse design methods, making it hard to judge relative performance gains.**
>
> As far as we are aware, there are no existing baselines that utilize text to guide the generation process of a design that resists physical deformation from an applied force. There are no datasets involved as we are using a differentiable physics simulator and a pre-trained text-image model to compute the loss terms.
>
> **W3. The manuscript’s novelty derives mainly from combining these known techniques in one pipeline (most core components—SIMP compliance optimization, CLIP-based text supervision and Hill-type density projection are adopted from earlier papers.)**
>
> The NeurIPS review guidlines state: "Does this work offer a novel combination of existing techniques, and is the reasoning behind this combination well-articulated?"
>
> This paper is the first to bridge two separate fields of research (text-image generation, and structural optimization). Its findings indicate TIDES, is well-suited for generating designs that resist structural deformation. This addresses a limitation of prior text generation works that are incapable of considering physical constraints and is an innovation toward realizing generated designs in life.
>
> **Hill-type density projection are adopted from earlier papers.**
>
> The Hill equation (equation 4) comes from bio-chemistry and has not to our knowledge been previously used in the context of structural optimization density projections. In standard structural optimization compliance alone is enough to drive the binary material distribution no extra steps are needed. The introduction of the visual loss leads to a competing design objective with shading encouraging a gray-scale distribution. We solve this by developing a Hill function with a steep slope to push binarization.
>
> **W2. It would be best to explain what the methodological contributions are**
>
> We developed a Hill function inspired activation (eq. 4) to restrict and push density values to 0 or 1, a compliance-based masking approach to ensure that the visual features offer structural support, a resampling approach to scale between different design resolutions, and a weighted loss to balance competing objectives. The key insight of the paper is that bridging a differentiable physics simulator with a pretrained text-image model allows text informed generation of strucutrally sound designs. This has implications for both the text to design and structural optimization.
>
> From the text to design side: The success with using a pretrained text-image model with a differentiable physics simulator indicates that it is not necessary to train a new text to design model from scratch on physics data to generate sound designs. Instead, bridging differentiable physics with pretrained text image models offers a promising new research direction towards realizing generated design in life.
>
> From the engineering/structural optimization side, we have shown that a visual text-image loss can be used to convey complex design goals and influence the features in the design while still respecting the underlying physical constraints. This is a promising new direction towards co-design, with text providing an intuitive means for conveying complex design goals to a physics-based optimizer and allowing for diverse design generation guided by text.

---

> > ### Comment · Reviewer_nNGQ · 2025-08-03
> >
> > Thank you for your detailed response. I appreciate the clarifications you have provided. But i still have some concerns that I believe need to be addressed for a more comprehensive evaluation of your method.
> >
> > 1. There is already some works on physics-constrained generation, including in both 2D and 3D domains. For example, works such as "Precise-Physics Driven Text-to-3D Generation" and "Atlas3D: Physically Constrained Self-Supporting Text-to-3D for Simulation and Fabrication" directly address the challenge of incorporating physical constraints into generative modeling. While your approach may differ in goals or implementation, I believe a more thorough discussion of these related works—including a comparison, even if qualitative—would help clarify your contributions and positioning. If direct comparisons are infeasible, it would still be beneficial to explain how your method differs and what specific advantages it offers.
> > 2. In addition, there are existing datasets that support 3D generation under physical constraints, such as DrivAerNet++. Evaluating your method on such datasets would significantly strengthen your claims. This is particularly important since 3D domains impose stricter and more complex physical constraints (e.g., CFD, structural forces), making them a better testbed to assess the generality and robustness of physics-aware generative models.
> >
> > I encourage the authors to consider addressing these points either in the final version or in future work.
> > I will keep my current score but may update based on further discussions with the reviewers and the AC.

---

> ### Author Response · Authors · 2025-08-04
>
> Thank you for your detailed response and providing additional context to your concerns. We believe there may be a misunderstanding of our approach in its placement to concurrent works (we tackle a different class of design problems), how it relates to data-based physics simulation approaches (a key benefit of our approach is that data is not required), and the appropriate baseline to compare against (we compare against the physics solution).
>
> During the past year (2024-2025) placing physical constraints on generative models has rapidly grown as an area of research with several works concurrent to this submission. To address this, we have added a new subsection to the related works and background section that summarizes these approaches and compares them with our approach.
>
> Our approach differs in that we tackle a class of design problems, structural optimization, rather than a specific design problem such as stability in "Atlas3D: Physically Constrained Self-Supporting Text-to-3D for Simulation and Fabrication". Structural optimization design problems cover diverse force and support application rather than a singular gravitational force. E.g. Appendix Figure 10, we present results from diverse design problems including horizontal force application, varied force application at different points, a suspend design that does not rest on the ground two-point cantilever beam, and many more throughout the paper and further a user can supply their own force and support placements and run our approach.
>
> Our approach does not rely on data and does not require seeding the starting design with a visual prior such as an image or a structure generated by a diffusion model. "Precise-Physics Driven Text-to-3D Generation" uses the output of a diffusion model as the starting design. This output is then fine tuned using a surrogate model trained to predict the physical performance from data. In our approach we can start from a solid block of material or empty canvas (we do not require a visual prior) and the physics simulator in our approach is a finite element solver wrapped in autograd, this allows us to directly backprop and optimize with regards to physics rather than towards an approximation. i.e. we jointly optimize the design from scratch rather than fine tuning an existing structure.
>
> Our approach is bench-marked against the physics solution. A key consideration in introducing a text-based design criterion is how much physical performance is trade-off to achieve the text goal. Concurrent works do not address this and only include comparisons against a visual baseline. In this work for each design problem as we are directly optimizing through the finite element solver we can directly compare against the physics solution (standard structural optimization solution) and for each design problem we present the physics only solution and find that there is a minimal trade-off in physical performance with designs generated by TIDES approaching the upper bound of designs generated from the standard structural optimization approach.
>
> DrivAerNet++ is concerned with a different design problem then the work under review. We are focused on structural optimization through minimizing compliance and serval of the methods we introduce in this work such as the compliance based masking approach are not applicable to the design problems present in DrivAerNet++. Further a key benefit of our approach is that data is not required as the finite element solver is wrapped in autograd allowing us to directly optimize towards the physic equations and compare directly against the physics solution.

---

### Official Review · Reviewer_75ov · 2025-07-02

**Clarity:** 3
**Significance:** 2
**Originality:** 2
**Rating:** 5
**Confidence:** 3

**Summary:**

This paper presents a structural optimization (topology optimization) technique for using text to add control of visual appearance. This is done through the addition of an augmented clip similarity loss and a compliance mask to remove disconnected components from a somewhat standard differentiable simulator based approach to topology optimization over a 2D pixel grid. Mapping of structural grids to the CLIP domain is achieved on the image side by interpreting density grids as greyscale RGB, and on the text side by biasing the text inputs towards black silhouettes by appending phrases such as "dark, black solid outline"

**Questions:**

- What, if any, tuning of parameters and CLIP text prompts (specifically the appended text like "black" and "silhouette") was needed to get these results, at what granularity was this done, and what were the final parameters used for the experiments? E.g., did the loss weights vary by image size, or by text target, or was there a universal set of values that worked well across experiments, and how big of an impact do these parameters have? Are there cases where one would want to use these weights to meaningfully control the designs?
- Please give technical details on how the compliance masking works sufficient to fully reproduce the experiments.
- How well is the density constraint achieved across the test examples, especially compared to the physics only baseline? Is there a useful tradeoff between aesthetic considerations and constraint satisfaction?

**Ethical Concerns:**

["NO or VERY MINOR ethics concerns only"]

**Final Justification:**

The rebuttal addressed my primary evaluation concern and provided a concise but sufficient description of the compliance masking details. I intend to maintain my recommendation.

Other reviewers have raised concerns over applicability to 3D and potential over-claming "physical soundness". I agree with the author's that generalization into "full 3D" is not claimed nor is necessary for publication (2/2.5D TO is useful in its own right). "Physically Sound" is potentially a very strong claim, but the paper does not make a firm definition of what it considers sound other than experimentally the design were 3D print-able. I agree that some of this language could be toned-down, or a brief explanation of what is meant by "soundness" added, but think this is not a major enough concern to reject the paper.

**Limitations:**

Yes

**Quality:**

3

**Strengths And Weaknesses:**

Strengths:
- A simple, straightforward, and clean architecture that (except for a few details) should be easy to reproduce
- Qualitative results look very good -- well aligned to input text prompts.

Weaknesses:
- Does not give details on the tuning necessary to achieve these results. Specifically, which weight parameters in the optimization loss were used, if these were consistent across trials or tuned for each one, and how much effort was used to do so. Similarly, was tuning (and possibly per-example tuning) required for the appended text strings to bias CLIP towards silhouette?
- Does not give details on how the compliance masking works
- Does not give any evaluation on how close to the target density design get (in absolute terms and compared to physics baseline). Since this is the only physics constraint (a soft constraint on density) that is explored, it is important to know how well this technique can balance constraints and optimization goals relative to a setup without the additional vision goal.

---

> ### Author Rebuttal · Authors · 2025-07-30
>
> Thank you for your review.
>
> **Q1. What, if any, tuning of parameters and CLIP text prompts (specifically the appended text like "black" and "silhouette") was needed to get these results, at what granularity was this done**
>
> We follow a fairly common prompting strategy of splitting the prompt into two parts the subject followed by the modifier, i.e. "subject, modifier" => "Eiffel tower, dark black outline". The subjects were selected to cover a wide array of topics to indicate robustness and range from the hello world of image generation "avacado arm chair" (Figure 3) to structural shapes such as "hexagon" (Figure 4) to artistic concepts such as "a giant sisphus pushing a stone boulder" (Figure 10) and many more. The modifier was selected to prompt in the binary region of the images CLIP was trained on. We arrived at the modifier of "dark black outline" as all designs should be solid dark black, not have shading and should have sufficient detail to capture the features. As outlines e.g. coloring books, are binary black and white and are detailed representations this was chosen as the modifier. In practice a user can supply any text prompt and as visual appearance is a goal there is some subjectivity on the user level on the modifier selection.
>
> **what were the final parameters used for the experiments?**
>
> In Appendix B, we provide all parameters used for each experiment.
>
> **did the loss weights vary by image size, or by text target, or was there a universal set of values that worked well across experiments, and how big of an impact do these parameters have?**
>
> The loss weights did not vary by image size or target text prompt. In Figure 3, the designs at varying resolution all utilize the same parameters. In Figures (1,2,4,5) the designs generated from different text prompts all use the same parameters. All results in the main body of the paper use the same parameters. The visual loss weight of 100 used in the main body of the paper was selected as it is fairly common in the literature to weight the clip score by 100. In Appendix Figure 10 for some of the design problems we increased the visual loss based to balance out an increase in the initial physics loss in some of the more complex design problems.
>
> **Are there cases where one would want to use these weights to meaningfully control the designs?**
>
> Yes, if aesthetic performance is valued over physical performance one could increase the visual weight to trade off physical performance. In the current work, our focus was on generating physically sound designs that utilized features specified by text and we opted for weights that preserved physical performance while utilizing visually recognizable features.
>
> **Q2. Please give technical details on how the compliance masking works sufficient to fully reproduce the experiments.**
>
> The compliance mask is generated by thresholding the element wise compliance returned from the physics simulator i.e. mask = log(compliance) >= threshold; threshold = -20. This returns a mask of zeros and ones where zero indicates a given pixel does not contribute structurally and one indicates it does. The mask is applied by multiplying by the design encoding (Image = mask*design), this is the first image augmentation performed. This ensures the CLIP loss only reflects pixels that are structurally contributing and prevents overfitting to the visual domain (i.e. prevents cheating the clip loss through creating floating features or non-structurally contributing features).
>
> **Q3. How well is the density constraint achieved across the test examples, especially compared to the physics only baseline?**
>
> The density constraint is achieved across all examples. Below we have included the mean and standard deviation in density across 30 trials for two different design problems and text prompts.
>
> | text prompt | mean  |   std
> | -------- | -------- | ------- |
> | a large $\mathbf{arch}$, dark black solid outline     |  0.301155    |  0.001447
> | a large $\mathbf{hexagon}$, dark black solid outline  |  0.300743    |  0.001627
> | a large $\mathbf{triangle}$, dark black solid outline |  0.301071    |  0.001910
> | baseline                                              |  0.301160    |  0.001846
> [suspended bridge design problem]
>
>
>
> | text prompt | mean  |   std
> | -------- | -------- | ------- |
> | a large $\mathbf{arch}$, dark black solid outline     |  0.300756      |   0.002440
> | a large $\mathbf{hexagon}$, dark black solid outline  |  0.300305      |   0.002358
> | a large $\mathbf{triangle}$, dark black solid outline |  0.299975      |   0.002504
> | baseline                                              |  0.299889      |   0.001756
> [hoop bridge design problem]

---

> > ### Comment · Reviewer_75ov · 2025-08-08
> >
> > Thank you for the additional details and clarifications!
> >
> > It appears I was mistaken about the CLIP text prompt varying between examples. My concerns and questions have been addressed, thank you!

---

### Official Review · Reviewer_ZiBG · 2025-07-03

**Clarity:** 3
**Significance:** 3
**Originality:** 2
**Rating:** 3
**Confidence:** 4

**Summary:**

The paper introduces TIDES, a method that jointly optimizes structural soundness and text-guided visual features in design. It combines a differentiable FEM simulator + a frozen CLIP model to generate 2.5D designs that are both physically viable and visually aligned but via CLIP guidance. Key contributions include a co-optimization framework, practical tricks like compliance masking and hill-based binarization, and some physical validation via 3D printed tests.

**Questions:**

- The core idea closely resembles Topology Optimization with Text-Guided Stylization Zhong et al. (2023), which also combines CLIP-based semantics with FEM optimization. Why is this not cited, and how does your method differ meaningfully?
- At higher resolutions (e.g. 512×512), designs become delicate. Are such fine structures actually manufacturable with  3D printing methods?
- The paper mentions 3D design, but the method operates on 2D grids with extrusion. Can it support native 3D FEM? If not, the terminology should be clarified.
- How sensitive is the method to prompt phrasing, e.g. “robot” vs. “bot”? Some discussion on this would clarify the controllability and robustness of your approach.

**Ethical Concerns:**

["NO or VERY MINOR ethics concerns only"]

**Final Justification:**

Given authors replies and esp the point around Zhong et al. (2023), I'd let the PC decide. The rebuttal has been okay, but didn't move the needle for me.

**Limitations:**

Yes, the authors include a limitations section (Appendix A) where they acknowledge the 2.5D constraint of their simulator, risks of misuse in generating real-world load-bearing structures, and the potential for bias amplification via CLIP. These points cover both technical and societal concerns sufficiently.

**Quality:**

3

**Strengths And Weaknesses:**

Strengths
- Quality: The method is technically sounds and includes good design choices like compliance masking and hill-based binarization. Experimental coverage is good and includes 3D-printed validation.
- Clarity: The paper is well-written with clear figures and thorough ablations that support the claims.
- Significance: Combines text-conditioned generative models with physics-aware structural design, and relevant in the AI-based computational design space.
- Originality: While the general idea is not entirely new, the implementation is clean, practical, and backed by real fabrication.

Weaknesses
- Originality: The core idea (CLIP + FEM optimization) overlaps with prior work (e.g. Topology Optimization with Text‑Guided Stylization, Zhong et al. 2023) and is not acknowledged or compared against.
- Scope: Designs are 2D with extrusion (~2.5D); while the paper references 3D printing and 3D design domains, the method itself does not support native 3D topology optimization or volumetric simulation
- Visual control: Visual constraints rely solely on text-to-CLIP embeddings, with no direct geometric control, which limits precision and style control.
- Limited generalization: The method is currently restricted to single-material, planar settings without manufacturability-aware constraints.

---

> ### Author Rebuttal · Authors · 2025-07-30
>
> Thank you for your review.
>
> **Q1. The core idea closely resembles Topology Optimization with Text-Guided Stylization Zhong et al. (2023), which also combines CLIP-based semantics with FEM optimization. Why is this not cited, and how does your method differ meaningfully?**
>
> The first version of the manuscript under review predates Zhong et al. (2023) and we were previously unaware of this work. We cannot provide additional details given the double-blind review policy but are happy to provide verification of this to the Area or Program Chair upon request. The current submission has been updated to address several of the challenges of linking CLIP with a finite element solver. The approach taken in Zhong can overfit to the visual domain and does not scale well to large designs spaces. These issues are not present in our approach we discuss below:
>
> Zhong utilizes RGB channels during the design generation process. This causes overfitting to the visual domain by allowing the model to improve the visual loss by painting over sections of the underlying structure rather than by shaping the physical structure itself. E.g. In Zhong Figure 12, the art deco building has off-white colored windows. These windows are not structural features and are painted on over solid material.
>
> Our work does not use RGB values, the focus is on generating a shared binary (material presence/absence, black or white pixel) distribution. The visual loss tends to favor generating artistically pleasant images that utilize shading rather than binary values. To solve this, we introduced a Hill-function into the design encoding to control material distribution and push pixel/density values towards 0 or 1. E.g. In Figure 3, we present designs generated by our approach for an art deco hotel. In contrast to Zhong the windows present in our designs all correspond to structural features (no material is present). In the designs generated by our approach the structural features are physically shaped by the text prompt, there is no RGB outlet for the model to cheat the loss by painting over structural features.
>
> Zhong does not scale well to large design spaces and can result in designs with features that do not offer structural support. Zhong utilizes the Connected Component Labeling (CCL) algorithm to remove unconnected features, this is computed at every optimization step. CCL scales poorly in computational time as the size of the design increases. In our work, we introduce a compliance based masking approach to remove unconnected features. As compliance is already computed for structural optimization there is no additional computational overhead required. Further compliance is physics based and allows the removal of features that are not offering direct structural support. In contrast CCL, can lead to connected features that do not offer structural support. E.g. In Zhong Figure 5b, after applying CCL there is a large amount of material that is connected to the main structure but offers no support, this material is hanging and attached by very thin connections that do not lend themselves well to 3D printing. In contrast, for a similar design problem in our work Figure 4, the support struts do not have these issues while displaying the text-informed features.
>
> We have added a summary of the above to the related works section and a citation to Zhong et al. 2023.
>
> **Q2. At higher resolutions (e.g. 512×512), designs become delicate. Are such fine structures actually manufacturable with 3D printing methods?**
>
> Yes, for delicate structures methods such as support bath based 3D printing or UV resin cure printers support printing designs with intricate details at very high resolution.
>
> **Q3. The paper mentions 3D design, but the method operates on 2D grids with extrusion. Can it support native 3D FEM? If not, the terminology should be clarified.**
>
> Yes, the Hill function and compliance based masking approach scale readily to 3D. For the visual loss a simple approach could be bounding the design in a cylinder and then sampling 2D slices to generate the images to compute the loss. To avoid confusion around 3D designs in the current work we state in the abstract: "2D beam designs that are extruded and 3D printed."
>
> **Q4. How sensitive is the method to prompt phrasing, e.g. “robot” vs. “bot”? Some discussion on this would clarify the controllability and robustness of your approach.**
>
> In practice, a user can provide any text prompt. In this work, we selected diverse prompts covering a wide array of topics to indicate robustness and range from the hello world of image generation "avacado arm chair" (Figure 3) to structural shapes such as "hexagon" and "arches" (Figure 4) to artistic concepts such as "a giant sisphus pushing a stone boulder" (Figure 10) and many more. For text prompts that cover very similar subjects e.g. “robot” vs. “bot”, the resulting design will be very similar. We have conducted a run for the subjects "robot" and "bot" the resulting designs display very similar robotic/bot features. Prompt engineering for CLIP has been well explored in the image generation literature, see section 2.2, and was not a focus of this paper.

---

> > ### Comment · Reviewer_ZiBG · 2025-08-05
> >
> > Overall, Q1: Given your comments, I cannot judge which paper came first, so I leave that to the Program Committee.
> > - It sounds like Zhong uses an RGB color channel, while your work uses a binary channel.
> > - Zhong operates in 3D, while your work is 2D/2.5D.
> > - I agree that 3D print validation is shown here (albeit on relatively simple structures), whereas Zhong does not include this.
> >
> > Q2: I am not sure. Having worked with 3D printing/UV resin curing, printability depends heavily on resolution and process. There is no study here to establish those limits. While you can claim 3D printability, it is hard to understand the ultimate goal if you are proposing a practical application.
> >
> > Q3: Thanks for the clarification.
> >
> > Q4: Your reply is noted.

---

> ### Author Response · Authors · 2025-08-05
>
> Thank you for your response. We provide two brief clarifications below.
>
> **Q1.**  We generally agree with the reviewer's comparison of this work to Zhong and would add the following bullet . We also note that use of RGB channels and CCL in Zhong both provide avenues for the design to overfit to the visual loss that are not present in our approach.
> - We use a physics based approach (compliance mask, no added computational cost) to remove both unconnected and non-structurally contributing features from the design whereas Zhong uses an image segmentation approach (Connected Component Labeling, scales poorly with design size) and only removes unconnected features from the design.
>
> **Q2.** In Q2 our intent was to indicate that the fine features present in the 512 x 512 designs are not unreasonable when compared to existing work on 3D printing intricate structures. E.g. "Recent innovations in interfacial strategies for DLP 3D printing process optimization" (2025) and "Fluid Bath-Assisted 3D Printing for Biomedical Applications: From Pre- to Postprinting Stages" (2021). Placing physical constraints on a generative text-image model is a relatively new area of research and is not yet at the level of practical application and we do not propose a practical application. We present evidence that this is a promising research direction towards the practical application of text-image models in physical design.

---

### Note · Authors · 2025-08-15

We offer a few final remarks and direct the AC and reviewers to where the remaining concerns/misunderstandings expressed in the reviewer response to the rebuttal are addressed. Thank you again for your reviews and rebuttal responses.

**Reviewer 75ov** and **Reviewer UJvm** indicated that our rebuttal addressed their concerns.

**Reviewer nNGQ** provided additional context in their response suggesting a possible misunderstanding of the baseline we evaluate against and how our work relates to concurrent works.

- The baseline we compare against in this work is structural optimization, the physics-based approach for generating a design. The physics baseline allows us to examine the trade off in physical performance caused by the introduction of a text/vision design goal. Concurrent works mentioned by the reviewers do not address this and only compare against designs generated from a text/vision baseline.

- This work cannot be directly compared with the concurrent works mentioned by the reviewer as they tackle different design problems from our work. For a detailed discussion see our rebuttal to reviewer UJvm and response to reviewer nNGQ.

**Reviewer fywP** requested the timing of the algorithm for different settings and expressed concerns around extruding a 2D design to 3D.

-  We have provided the specific timings for running our algorithm at different resolutions in our response to reviewer fywP. The slow step is the physics simulator and this is shared by both our approach and the structural optimization baseline.

- The 3D printed and evaluated designs in this work correspond to a widely used structural optimization baseline (MBB-beam/3-point-bending) in which 2D designs are extruded and 3D printed. For recent examples of this baseline see our response to reviewer fywP.

**Reviewer ZiBG** expressed concerns around the 3D printability of higher resolution designs.

- We have provided two examples of recent work that print intricate high resolution designs that are more detailed than our designs using support bath based printers and UV resign cure printers, see our response to reviewer ZiBG. We also clarify that 3D printability was not a focus of the current work.

---

### Decision · Program_Chairs · 2025-09-17

**Decision:**

Accept (poster)

**Comment:**

The paper proposes a method that generates 2D designs by jointly optimizing a text-to-image visual loss (measured through CLIP), a material cost, and a physics-based loss (measured through a differentiable physics simulator). The main strengths of the paper are: (a) its novelty based on combining SIMP compliance optimization, CLIP-based text supervision, and Hill-type density projection to achieve physically sound designs; and (b) a generic framework that can accommodate various 2D forces (beyond gravity, as supported by prior work). There were a few identified weaknesses, such as: (a) the method is limited to 2D structures, (b) many hyperparameters may require hand-tuning for different designs, and (c) similarities to prior work that are not fully discussed, specifically Zhong et al., Topology Optimization with Text-Guided Stylization, Structural and Multidisciplinary Optimization, 2023.

The paper received one accept, one reject (nNGQ), two borderline accepts, and one review (ZiBG) where the reviewer deferred the decision to the "PC". The AC carefully read both the submission and Zhong et al. (2023), and agrees with the authors that the latter can overfit to the visual domain due to its style optimization loss, whereas the proposed method is more technically sound. Reviewer nNGQ argued that the paper does not compare with "important related works on physics-constrained generation". However, the AC concurs with the authors’ view that the works cited address different design problems (e.g., stability under a singular gravitational force). Since no reviewer provided a strong and detailed justification for rejecting the paper solely based on prior work addressing different goals, the AC recommends acceptance.

That said, the authors are strongly encouraged to expand their discussion of all prior and concurrent work. In particular, they are strongly encouraged to include a figure comparing with Zhong et al. (2023) -- such as the art deco building example mentioned in the rebuttal where Zhong et al. fails. More broadly, the authors are encouraged to incorporate all clarifications from the rebuttal and the discussion into the final version of the paper and supplementary material!